# Detecting volcanic sulfur dioxide plumes in the Northern Hemisphere using the Brewer spectrophotometers, other networks, and satellite observations

Christos S. Zerefos[1,2,3,4], Kostas Eleftheratos[2,5], John Kapsomenakis[1], Stavros Solomos[6], Antje Inness[7], Dimitris Balis[8], Alberto Redondas[9], Henk Eskes[10], Marc Allaart[10], Vassilis Amiridis[6], Arne Dahlback[11], Veerle De Bock[12], Henri Diémoz[13], Ronny Engelmann[14], Paul Eriksen[15], Vitali Fioletov[16], Julian Gröbner[17], Anu Heikkilä[18], Irina Petropavlovskikh[19], Janusz Jarosławski[20], Weine Josefsson[21], Tomi Karppinen[22], Ulf Köhler[23], Charoula Meleti[8], Christos Repapis[4], John Rimmer[24], Vladimir Savinykh[25], Vadim Shirotov[26], Anna Maria Siani[27], Andrew R. D. Smedley[24], Martin Stanek[28], René Stübi[29]

[1]Research Centre for Atmospheric Physics and Climatology, Academy of Athens, Athens, Greece
[2]Biomedical Research Foundation, Academy of Athens, Athens, Greece
[3]Navarino Environmental Observatory (N.E.O.), Messinia, Greece
[4]Mariolopoulos-Kanaginis Foundation for the Environmental Sciences, Athens, Greece
[5]Faculty of Geology and Geoenvironment, National and Kapodistrian University of Athens, Greece
[6]Institute for Astronomy, Astrophysics, Space Applications and Remote Sensing (IAASARS), National Observatory of Athens, Athens, Greece
[7]European Centre for Medium-Range Weather Forecasts (ECMWF), Reading, UK
[8]Department of Physics, Aristotle University of Thessaloniki, Thessaloniki, Greece
[9]Izaña Atmospheric Research Center, AEMET, Tenerife, Canary Islands, Spain
[10]Royal Netherlands Meteorological Institute (KNMI), De Bilt, the Netherlands
[11]Department of Physics, University of Oslo, Oslo, Norway
[12]Royal Meteorological Institute of Belgium, Brussels, Belgium
[13]ARPA Valle d'Aosta, Saint-Christophe, Italy
[14]Leibniz Institute for Tropospheric Research, Leibniz, Germany
[15]Danish Meteorological Institute, Copenhagen, Denmark
[16]Environment and Climate Change Canada, Toronto, Canada
[17]PMOD/WRC, Davos Dorf, Switzerland
[18]Climate Change Unit, Finnish Meteorological Institute, Helsinki, Finland
[19]Cooperative Institute for Research in Environmental Sciences, University of Colorado, Boulder, CO USA
[20]Institute of Geophysics, Polish Academy of Sciences, Warsaw, Poland
[21]Swedish Meteorological and Hydrological Institute, Norrköping, Sweden
[22]Arctic Research Centre, Finnish Meteorological Institute, Sodankylä, Finland
[23]DWD, Meteorological Observatory Hohenpeissenberg, Germany
[24]Centre for Atmospheric Science, School of Earth, Atmospheric and Environmental Sciences, University of Manchester, Manchester M13 9PL, UK
[25]A.M. Obukhov Institute of Atmospheric Physics, Kislovodsk, Russia
[26]Institute of Experimental Meteorology, Obninsk, Russia
[27]Department of Physics, Sapienza – University of Rome, Rome, Italy
[28]Solar and Ozone Observatory, Czech Hydrometeorological Institute, Hradec Kralove, Czech Republic
[29]Federal Office of Meteorology and Climatology, MeteoSwiss, Payerne, Switzerland

*Correspondence to*: Christos S. Zerefos (zerefos@geol.uoa.gr)

**Abstract.** This study examines the adequacy of the existing Brewer network to supplement other networks from ground and space to detect $SO_2$ plumes of volcanic origin. It was found that large volcanic eruptions of the last decade in the Northern Hemisphere have a positive columnar $SO_2$ signal seen by the Brewer instruments located under the plume. It is shown that a few days after the eruption the Brewer instrument is capable of detecting significant columnar $SO_2$ increases, exceeding on the average 2 DU relative to an unperturbed pre-volcanic 10-day baseline, with a mean close to zero and $\sigma = 0.46$, as calculated from the 32 Brewer stations under study.

Intercomparisons with independent measurements from ground and space as well as theoretical calculations corroborate the capability of the Brewer network to detect volcanic plumes. For instance, the comparison with OMI and GOME-2 $SO_2$ space-borne retrievals shows statistically significant agreement between the Brewer network data and the collocated satellite overpasses in the case of Kasatochi eruption. Unfortunately, due to sparsity of satellite data the significant positive departures seen in the Brewer and other ground networks following Eyjafjallajökull, Bárðarbunga and Nabro eruptions could not be statistically confirmed by the data from satellite overpasses. A model exercise from the MACC (Monitoring Atmospheric Composition and Climate) project shows that the large increases of $SO_2$ over Europe following the Bárðarbunga eruption in Iceland were not caused by local pollution sources or ship emissions but were clearly linked to the volcanic eruption. Sulfur dioxide positive departures in Europe following Bárðarbunga could be traced by other networks from the free troposphere down to the surface (AirBase and EARLINET). We propose that by combining Brewer data with that from other networks and satellites, a useful tool aided by trajectory analyses and modelling could be created which can be used also to forecast high $SO_2$ values both at ground level and in air flight corridors following future eruptions.

**1 Introduction**

Volcanic eruptions are an important source of natural emissions of sulfur dioxide ($SO_2$) into the troposphere and the stratosphere. Ash particles and gases injected into the atmosphere by large volcanic eruptions can affect solar radiation and climate (e.g. Robock, 2000), air quality (e.g. Schmidt et al., 2015) and may also impact local environments (e.g. Durant et al., 2010). Volcanic emissions (e.g. ash and $SO_2$) can reach different heights in the atmosphere and can be transported in different directions (e.g. Prata et al., 2010). Thomas and Prata (2011) have shown that the eruption can be divided into an initial ash rich phase, a lower intensity middle phase and a final phase where considerably greater quantities of both ash and $SO_2$ are released which in the case of ash can result in air travel disruptions (e.g. Flentje et al., 2010). These effects make the ash and $SO_2$ in volcanic plumes important parameters to be studied, monitored and forecasted on small and larger spatial scales. Our study focuses on volcanic columnar $SO_2$ amounts because of the existence of the rather continuous set of direct sun measurements with the Brewer network.

Measurements of $SO_2$ are important for tracking and assessing impacts of emissions from pollution sources and in quantifying natural $SO_2$ emissions by volcanoes. Pollution sources typically result in a few Dobson Units (DU, 1 DU = $2.69 \cdot 10^{26}$ molec$\cdot$km$^{-2}$) increases of column $SO_2$ amounts unless observations are made near a source. The Brewer network is useful for plume tracking because it can track $SO_2$ columnar amounts from a large number of stations and wide geographical extent. The primary application of the ground-based Brewer spectrophotometer is to measure total ozone column by using UV spectrophotometry. Direct sunlight intensities are measured at five wavelengths (between 306 and 320 nm; see also Sect. 2.1) to simultaneously calculate ozone and $SO_2$ column integrals (Kerr et al., 1980). These instruments have been used extensively to monitor stratospheric ozone (e.g. WMO Scientific Assessment of Ozone Depletion reports 2011, 2014) and have a long history of studying atmospheric $SO_2$ columns (e.g. De Backer and De Muer, 1991; Bais et al., 1993; Fioletov et al., 1998; Zerefos et al., 2000; Zerefos et al., 2009; Ialongo et al., 2015). Ground-based measurements of

atmospheric $SO_2$ using the Brewer instrument have played an important role in the development and validation of satellite-based $SO_2$ measurements (Schaefer et al., 1997; Spinei et al., 2010; Rix et al., 2012; Ialongo et al., 2015) used primarily for detecting and tracking volcanic emissions. Since the Brewer instruments are located at stationary ground-based monitoring sites, a volcanic plume of $SO_2$ must pass over the site if useful data are to be obtained. Validation of satellite measurements by the Brewer instrument also requires that a satellite overpass is available when the plume is over the ground based site (Kerr, 2010).

There have been various initiatives during recent years that used satellite measurements of $SO_2$ to monitor volcanic eruptions in support of aviation safety, e.g. ESA's Support to Aviation Control Service (SACS) (Brenot et al., 2014). These initiatives together with modelling forecasting tools provide valuable information to the established Volcanic Ash Advisory Centers (VAAC). Satellite $SO_2$ data have been available in the past from various satellite instruments (e.g. GOME, SCIAMACHY). Currently operational data are available from UV measurements (e.g. GOME-2 (Global Ozone Monitoring Experiment-2), OMI (Ozone Monitoring Instrument) and OMPS (Ozone Mapping Profiler Suite)) and from infrared measurements (e.g. IASI (Infrared Atmospheric Sounding Interferometer) and AIRS (Atmospheric Infrared Sounder)).

In the present work we investigate the efficiency of the existing Brewer network in the Northern Hemisphere to detect volcanic $SO_2$ plumes during the past decade. The main focus is to show the sensitivity of the Brewer network in detecting $SO_2$ plumes of volcanic origin in synergy with other ground based observations, satellite data and dynamic transport calculations. The Brewer spectroradiometric measurements are compared to collocated satellite measurements from OMI and GOME-2 as described in the next paragraph. This paper did not include analyses of the $SO_2$ measurements from IASI and AIRS since both instruments are IR spectroradiometers. We compared Brewer measurements against the OMI and GOME-2 data that are derived using information from differential optical absorption in the UV spectrum, which is also at the base of the Brewer measurement methodology. In the case of Brewer-IASI or Brewer-AIRS comparison we would also have to consider differences in the spectroscopy and the corresponding retrieval algorithm concepts, which would require further analysis which is beyond the scope of this paper.

Table 1 lists in chronological order all major volcanic eruptions in the Northern Hemisphere between 2005-2015 with volcanic explosivity scale index (VEI) of at least 4 (Newhall and Self, 1982; Robock et al., 2000; Zerefos et al., 2014). The study also provides a separate analysis for the Bárðarbunga eruption, which although not rated 4 has been already studied with the Brewer at Sodankylä by Ialongo et al. (2015).

As seen from Table 1, chronologically, the first case was the volcanic eruption at Mount Okmok, Alaska (53.43$^o$ N, 168.13$^o$ W, 1073 m above sea level (asl), 12 July 2008, Prata et al., 2010) followed by the Kasatochi eruption, Alaska (52.17$^o$ N, 175.51$^o$ W, 300 m asl, 7-8 August 2008, e.g., Kristiansen et al., 2010; Krotkov et al., 2010; Waythomas et al., 2010) which was detected over large areas of the Northern Hemisphere. Okmok and Kasatochi volcanoes in Alaska erupted a short time span of less than a month and therefore we decided to study the evolution of the Brewer $SO_2$ columnar measurements following the latter volcanic eruption (Kasatochi). The third eruption took place at Sarychev in Russia (48.1$^o$ N, 153.2$^o$ E, 1496 m asl, 12-17 June 2009, Haywood et al.,

2010). The evolution of the $SO_2$ volcanic plume from Sarychev was mostly observed over the North Pacific, North America and North Atlantic (Haywood et al., 2010). There was only one North American Brewer station (Saturna Island) in the path of the plume from Sarychev eruption. The record shows $SO_2$ columns of 8.6 DU detected on 19 June 2009 and 3.7 DU on 20 June 2009. This volcanic eruption is not investigated any further in this paper. The next eruption on the list, Eyjafjallajökull in Iceland (63.63$^o$ N, 19.62$^o$ W, 1666 m asl, from 14 April to 23 May 2010), resulted in interruption of the air traffic over NW Europe (e.g. Flemming and Inness, 2013). The fifth eruption Grímsvötn 2011 (64.42$^o$ N, 17.33$^o$ W, 1725 m asl, 21 May 2011) was studied by Flemming and Inness (2013), and by Moxnes et al. (2014). This eruption provided an interesting example of a clear separation of the volcanic $SO_2$ plume (transported mostly northwestward) while the fine ash was transported mostly southeastward. Unfortunately the volcanic plume did not overpass any Brewer station and therefore we do not include any results post Grímsvötn eruption. The sixth eruption recorded features the Nabro in Africa (13.37$^o$ N, 41.70$^o$ E, 2218 m asl) that occurred on 12-13 June 2011 (e.g., Bourassa et al., 2012; Sawamura et al., 2012; Clarisse et al., 2014). We present here a case study that described detection of the Nabro volcanic $SO_2$ plume over ground based stations. The plume was clearly detected by the Brewer instrument over Izaña (and poorly from space), then over Taiwan by both Brewer and satellite instruments, and finally at Mauna Loa, Hawaii (mostly by the Brewer instrument). The seventh eruption was Tolbachik, Russia (55.83$^o$ N, 160.33$^o$ E, 3.611 m asl) on 27 November 2012 (e.g. Telling et al., 2015). As in the case of Grímsvötn, the plume has not passed over any Brewer station that was verified by trajectory analysis. The next eruption on the list is the volcanic eruption from Bárðarbunga, Iceland (64.64$^o$ N, 17.56$^o$ W, 2005 m asl) that was observed between 31 August 2014 and 28 February 2015 (e.g. Schmidt et al., 2015). This last eruption, although not yet rated on the VEI scale, has been extensively studied in view of the observed increased $SO_2$ concentrations that have been observed all the way through troposphere and reaching down to the surface in Europe (Ialongo et al., 2015; Schmidt et al., 2015).

The capability of the Brewer network to measure columnar $SO_2$ amounts above the local air pollution levels is also presented and discussed. The qualitative evidence that the plume can be detected in many single cases by the Brewer network has been quantitatively tested by calculating correlation coefficients with collocated satellite data. Only in the case of Kasatochi 2008 eruption it was possible to test the sensitivity of $SO_2$ abundance measured by the Brewers and from space. Correlations between the Brewer and collocated satellite $SO_2$ data from the Aura OMI and GOME-2 are presented in the section 3 where the correlation coefficients were found to be statistically significant at a confidence level of 99%. For the other eruptions unfortunately due to the sparsity of satellite data no firm conclusions can be drawn as discussed in section 3.

The paper is structured in the following order, Section 2 describes the data sources and the methods of analysis of the columnar $SO_2$ measurements by the Brewer spectrophotometers (hereinafter simply referred to as the "Brewers"). Section 3 presents the analysis of the Brewer measurements during four of the volcanic eruptions listed in Table 1, along with satellite data and dynamic volcanic plume transport simulations. The conclusions are provided in Section 4.

**2 Data and methods**

## 2.1 Ground based data

Sulfur dioxide in the atmosphere can be measured from ground-based instruments, by instrumentation onboard the spacecraft and can be estimated with help of models. The Brewer is an automated, diffraction-grating spectrophotometer that provides observations of the sun's intensity in the near UV range. The spectrophotometer measures the intensity of radiation in the ultraviolet absorption spectrum of ozone at five wavelengths (306.3 nm, 310.1 nm, 313.5 nm, 316.8 nm and 320.1 nm) with a resolution of 0.6 nm. These data are used to derive the total ozone column (Kerr et al., 1980). Because sulfur dioxide has strong and variable absorption in this spectral region, the Brewer spectrophotometer has additionally been proposed to derive $SO_2$ columns (Kerr et al., 1980). About two hundred Brewer spectrophotometers around the world contribute high-precision ozone data to the global ozone monitoring network (Kumharn et al., 2012). The existing Brewer network also delivers frequent $SO_2$ columnar measurements as well, which can be used for analyses, but with caution. This is because the signal to noise ratio for the $SO_2$ absorption is usually quite low and therefore well calibrated instruments are required to monitor nominal $SO_2$ columnar amounts (Koukouli et al., 2014). Details on the method with which $SO_2$ is measured by the Brewer spectrophotometer can be found in Kerr et al. (1980; 1985; 1988) and De Backer and De Muer (1991). According to Fioletov et al. (2016) the uncertainty of the Brewer direct sun (DS) $SO_2$ measurements is about 1 DU and is typically insufficient for air quality applications. A more accurate method (with an uncertainty as low as 0.13 DU) based on Brewer "group-scan" spectral direct sun radiation measurements at 45 wavelengths from 306 to 324 nm was developed (Kerr, 2002), but not implemented for routine operations due to its complexity (Fioletov et al., 2016). Although the Brewer instrument has difficulties in detecting low columnar $SO_2$ concentrations, in extreme cases, such as volcanic eruptions, the $SO_2$ levels typically rise well above the instrumental noise and can be identified with the Brewer instrument as shown in this paper and in Fioletov et al. (1998).

Before proceeding to the analysis of Brewer measurements, the methodology to derive columnar $SO_2$ is first presented. To determine ozone and $SO_2$ column amounts, the measured raw photon counts at the five operational channels in the Brewer instrument are converted to radiation intensity. The Beer-Lambert absorption law is applied at each wavelength λ, and the measured intensity of direct sunlight is given by the formula:

$$\log I_\lambda = \log I_{0\lambda} - \beta_\lambda \mu_R - \delta_\lambda \mu_p - \alpha_\lambda O_3 \mu - \sigma_\lambda SO_2 \mu \qquad (1)$$

where $I_\lambda$ is the measured radiation intensity at wavelength λ, $I_{0\lambda}$ is the measured extra-terrestrial spectrally resolved intensity at λ, $\beta_\lambda$ is the Rayleigh scattering coefficient at λ, $\delta_\lambda$ is the particulate scattering coefficient at λ, $\alpha_\lambda$ is the ozone absorption coefficient (cm$^2$/molecules) at λ, $O_3$ is the total ozone column (molecules/cm$^2$), $\sigma_\lambda$ is the $SO_2$ absorption coefficient at λ, $SO_2$ is the column amount of sulfur dioxide, $\mu_R$, $\mu_p$ and $\mu$ are the optical path lengths (air masses) corresponding to the effective heights of molecules, particles, and ozone respectively.

According to the Brewer retrieval algorithm, the following ratios are formed:

$$F = F_0 - \Delta\beta \mu_R - \Delta\alpha O_3 \mu \qquad (2)$$

and

$$F' = F_0' - \Delta \beta' \mu_R - \Delta \alpha' O_3 \mu - \Delta \sigma' SO_2 \mu \tag{3}$$

where $F$ is the weighted ratio of direct sun measurements at 4 (or 6 for double Brewer) spectral channels, $F = logI_2 - 0.5\ logI_3 - 2.2\ logI_4 + 1.7\ logI_5$ , $F_0$, $\Delta\beta$, and $\Delta\alpha$ are the same linear combinations for $\mathbf{log}I_{0\lambda}$, $\boldsymbol{\beta_\lambda}$, and $\boldsymbol{\alpha_\lambda}$. The $\boldsymbol{F'}$ is the SO$_2$ ratio, $F' = logI_1 - 4.2\ logI_4 + 3.2\ logI_5$ and $\boldsymbol{F_0'}$, $\boldsymbol{\Delta\beta'}$, $\boldsymbol{\Delta\alpha'}$ and $\boldsymbol{\Delta\sigma'}$ the corresponding linear combinations for $\mathbf{log}I_{0\lambda}$, $\boldsymbol{\beta_\lambda}$, $\boldsymbol{\alpha_\lambda}$, $\boldsymbol{\sigma_\lambda}$. Both of these functions have weights which eliminate the effects of particulate scattering, while the function $F$ is weighted to remove SO$_2$ absorption effects as well. The extra-terrestrial constants $\boldsymbol{F_0}$ and $\boldsymbol{F_0'}$ are determined from a long series of intercomparison measurements as well as zero air mass ($\boldsymbol{\mu}$) extrapolations.

The total ozone column is determined by the formula

$$O_3 = \frac{F_0 - F - \Delta\beta\mu_R}{\Delta\alpha\ \mu} \tag{4}$$

and the SO$_2$ by the formula

$$SO_2 = \frac{1}{A}\left(\frac{F_0' - F' - \Delta\beta'\mu_R}{\Delta\alpha'\ \mu} - O_3\right) \tag{5}$$

where $A$ is the ratio of the SO$_2$ absorption coefficient to the O$_3$ absorption coefficient, $A = 2.44$.

From the above described operational Brewer algorithm it is evident that the estimation of columnar SO$_2$ is the result of the difference between two columnar terms (O$_3$ + SO$_2$) and O$_3$. Both terms have uncertainties (weighting functions, calibrations, random errors, systematic errors). Systematic negative values could be the result of a systematic offset in the measurements that can be related to the calibration of the instrument (usually optimized only for the ozone measurements). Randomly varying positive and negative values around zero, suggest that the signal of SO$_2$ is small (and thus the difference of two terms should be close to zero) but since both terms have uncertainties, negative values are possible indicating that the amount of SO$_2$ in the atmosphere is below the detection limit of the instrument and could be considered as noise. In this work we have repeated our analysis excluding the negative values and the results remained the same i.e. a positive increase after a major volcanic eruption was confirmed as described in the following sections. Finally, we need to point out that perturbations by ash present in the volcanic plumes have been shown not to affect the Brewer SO$_2$ measurements. This is based on the result of Pappalardo et al., 2013 paper based on EARLINET observations following the Eyjafjallajökull eruption in which they found that the Ångström exponent of the volcanic ash optical depth is close to zero. This indicates that the effect of ash in the UV and visible region on the aerosol extinction is almost independent from wavelength. The Brewer SO$_2$ measurements taken in a narrow wavelength band in the UV are therefore not expected to be influenced by the presence of volcanic ash considering the weights already applied in the operational Brewer algorithm.

In this study we analysed twenty three stations located in Europe, six Brewer stations in Canada, two in the USA and one in Taiwan, whose geographical positions are shown in Figure 1. $SO_2$ measurements were averaged over a large number of instruments and datasets during periods following volcanic eruptions. Random errors in the measurements of individual Brewer stations are reduced significantly by the averaging processes to calculate regional means.

Daily $SO_2$ columns at Churchill, Goose, Edmonton, Regina, Saturna Island and Toronto in Canada, Taipei in Taiwan, Boulder and Mauna Loa in the US were obtained from the World Ozone and Ultraviolet Radiation Data Centre (WOUDC; http://www.woudc.org/) and the NOAA-EPA Brewer Spectrophotometer UV and Ozone Network (NEUBrew; http://www.esrl.noaa.gov/gmd/grad/neubrew/). The data have been checked for quality assurance/quality control by the individual data providers. It is important to note the participation of the most of the European Brewer data providers in a recent EU COST Action (EUBREWNET, http://www.eubrewnet.org/cost1207/) programme. Its focus is at establishing a coherent network of European Brewer Spectrophotometer monitoring stations in order to harmonise operations and develop approaches, practices and protocols to achieve consistency in quality control, quality assurance and coordinated operations.

In our analysis only direct sun (DS) measurements satisfying the following criteria have been used: a Brewer DS measurement was included if and only if for every measurement cycle of 5 sets of measurements (from which also total columnar ozone is derived) the standard deviation of $O_3$ and $SO_2$ was less than 2.5 DU, the total columnar ozone was between 250 DU and 450 DU, and the solar zenith angle was less than 73.5 degrees. To exclude erratic data of $SO_2$ from our analysis, values exceeding $\pm 6\sigma$ of the mean of all $SO_2$ individual Brewer measurements were considered erroneous and were not included in the calculations. Therefore the range of analysed values were limited to a maximum of $\pm$ 35 DU for an individual measurement (i.e. $6\sigma$, with $\sigma$ being equal to 5.8 as estimated from all available sub-daily $SO_2$ values). Then we calculated daily $SO_2$ columns at each station only if at least three individual measurements passed these criteria for each day. Brewers are useful because they provide more than one observation per day. For plumes which change rapidly, more than one observation per day would be useful, especially to complement satellites which typically have just one local overpass.

Daily sulfur dioxide ($SO_2$) columns were analysed in four bimonthly periods, namely August-September 2008, April-May 2010, June-July 2011 and September-October 2014, which include the volcanic eruptions of Kasatochi (2008), Eyjafjallajökull (2010), Nabro (2011) and Bárðarbunga (2014), respectively. For the case of Kasatochi, Eyjafjallajökull and Bárðarbunga we analysed daily $SO_2$ columns at 30 sites (listed in Table 2), while for the case of Nabro, whose impact was mostly seen over low latitudes in the N.H. (e.g., Bourassa et al., 2012), we analysed $SO_2$ columns at three low latitude sites in the Northern Hemisphere, namely Izaña, Mauna Loa and Taipei.

Only for the case of the Bárðarbunga eruption in 2014, the columnar $SO_2$ measurements over Europe were also compared with surface $SO_2$ measurements from ground based European stations. The surface $SO_2$ data were

obtained from the European Environment Agency databases (AirBase; http://www.eea.europa.eu/data-and-maps/data/aqereporting-1#tab-european-data) covering the bimonthly period September-October 2014. Only rural background stations, i.e stations in class 1-2 according to the Joly-Peuch classification methodology for the surface sulfur dioxide (Joly and Peuch, 2012), located at a distance of less than 150 km from the nearest Brewer station, were used in the analysis. A total of 7 stations in Europe (see Table 3) fulfilled the above mentioned criteria and were included in the current analysis. Observed data from the AirBase network were available in hourly resolution, from which we calculated daily surface $SO_2$ values. We note here that $SO_2$ in the troposphere over Western Europe is very low (e.g. Zerefos et al., 2009; Wild, 2012) and therefore plumes from volcanic eruptions are easy to detect against a lower background level.

## 2.2 Satellite Data

The columnar $SO_2$ records from remote sensing spectrophotometers over Europe, Canada, USA and Taiwan were compared with space-borne measurements from a) the Ozone Monitoring Instrument (OMI) aboard the EOS-Aura (e.g. Ialongo et al., 2015) satellite and b) the Global Ozone Monitoring Experiment-2 (GOME-2) aboard the MetOp-A (e.g. Rix et al., 2009) satellite. We use MetOp-A instead of MetOp-B because it covers a longer time period. Both OMI and GOME-2 satellite $SO_2$ data products were downloaded from the Aura Validation Data Center (AVDC) (available from: http://avdc.gsfc.nasa.gov/index.php?site=245276100). GOME-2 level 2 overpass data have been processed with the GOME Data Processor (GDP) version 4.7. We analysed station overpass data for the various mid-latitude stations listed in Table 2 and for the low latitude stations at Mauna Loa, Izaña and Taipei. The available OMI version 1.2.0 overpass (collection 3) data analysed in this study include pixels within 50 km radius from the nearest Brewer site and is not affected by OMI row anomalies. The available GOME-2 level 2 overpass data include pixels within 100 km radius from the Brewer sites.

For the case of OMI, the $SO_2$ data are provided from October 2004 to the present. There are four $SO_2$ products: (1) the Planetary Boundary Layer $SO_2$ column (PBL), corresponding to a centre of mass altitude (CMA) at 0.9 km, (2) the lower tropospheric $SO_2$ column (TRL) corresponding to CMA of 2.5 km, (3) the middle tropospheric $SO_2$ column (TRM), usually produced by volcanic degassing, corresponding to CMA of 7.5 km, and (4) the upper tropospheric and stratospheric $SO_2$ column (STL), usually produced by explosive volcanic eruptions, corresponding to CMA of 17 km. Details on OMI $SO_2$ columns can be found in various studies (Levelt et al., 2006; Yang et al., 2007; Fioletov et al., 2011; McLinden et al., 2012; Fioletov et al., 2013; Li et al., 2013; Ialongo et al., 2015). In this study, we made use of the product for the middle tropospheric $SO_2$ column (TRM) following the recommendation that the TRM retrievals should be used for volcanic degassing at all altitudes, because the PBL retrievals are restricted to optimal viewing conditions and TRL data are overestimated for high altitude emissions (>3km) (Ialongo et al., 2015). Also, we analysed the STL data which are intended for use with explosive volcanic eruptions where the volcanic cloud is placed in the upper troposphere / stratosphere. The standard deviation of TRM retrievals in background areas is reported to be about 0.3 DU in low and mid-latitudes and about 0.2 DU for the STL retrievals. This is similar to the standard deviations (indicative of typical uncertainties of the measurements) that we find for the TRM and STL retrievals in the four bimonthly periods under this study. For the best data quality, we used data from the scenes near the centre of the OMI swath (rows

4-54) as recommended by Ialongo et al. (2015) who found that data from the edges of the swath tend to have greater noise.

For GOME-2, we analysed the total $SO_2$ columns from April 2007 to the present. The standard deviation found in our study for the GOME-2 retrievals is the order of 0.4 DU. We analysed satellite $SO_2$ measurements whenever $O_3$ column was between 250 and 450 DU and solar zenith angle was less than 73.5 degrees. We used $SO_2$ data defined as having a cloud radiance fraction (across each pixel) less than 50%, as they were found to have smaller standard deviation than all sky data. Moreover, a range of $SO_2$ values between -35 and 35 DU was used to screen for outliers. In cases of multiple daily data matched to the station overpass, all available measurements within a radius of 50 (100) km from the Brewer site in the case of OMI (GOME-2) are averaged.

Finally, both for the satellite and the Brewer data we have considered that during a ten-day period prior to any eruption both the surface and the satellite data sets represent a baseline reference from which subsequent departures after the eruption should be tested as to their significance. Therefore, we calculated averages and standard deviations ($\sigma$) of departures from the unperturbed pre-volcanic period, for the three studied periods of volcanic importance at each station, only if at least 25 daily values were available. The bimonthly averages for each station in the examined periods are presented in Table 4a. Table 4b shows the mean and standard error ($\sigma/\sqrt{N}$) of all bimonthly averages in each studied volcanic period. Averaging the departures from the pre-volcanic baseline for all Brewer stations and for all bimonthly periods gives a mean $SO_2$ columnar departure of $0.10 \pm 0.03$ DU. This estimate is on the same order of magnitude as the corresponding statistics for OMI (TRM) $SO_2$ column departures ($0.05 \pm 0.02$ DU), OMI (STL) ($0.04 \pm 0.01$ DU) and that measured by GOME-2 ($0.09 \pm 0.02$ DU). The standard deviation of the bimonthly averages relative to their baselines, which was calculated from a large sample of data, was taken here as an approximation of the typical uncertainties in the columnar $SO_2$ measurements performed by the group of Brewers, OMI and GOME-2 instruments following volcanic eruptions.

## 2.3 Modelling tools

Dispersion of volcanic emissions is simulated with the Lagrangian transport model FLEXPART (Stohl et al., 2005; Brioude et al., 2013). The model is driven by hourly meteorological fields from the Weather Research and Forecasting (WRF) atmospheric model (Skamarock et al., 2008) at a horizontal resolution of 45×45 km. The initial and boundary conditions for the WRF model are taken from the National Center for Environmental Prediction (NCEP) final analysis (FNL) dataset at 1°×1° resolution. The sea surface temperature (SST) is initialised from the NCEP 1°×1° analysis. A total of 40,000 tracer particles are assumed for each release in FLEXPART simulations. The use of 1-hourly WRF meteorological fields at 45x45 km spatial resolution allows a more detailed representation of the volcanic plume dispersion but implies also a significant increase in computational time. To overcome this computational time cost, source-receptor relationships between station measurements and volcanic activity are also analysed with the use of HYSPLIT model trajectories (Stein et al., 2015) of long range transport driven by the 3-hourly meteorological dataset Global Data Assimilation System (GDAS) at a resolution of 1°×1°.

## 3 Results and discussion

### 3.1 The 2014 Bárðarbunga case

Bárðarbunga was continuously active during September – October 2014 but it was only during 18-26 September when meteorological conditions favoured transport towards Europe as shown by back trajectories analyses. A detailed description of the transport of Bárðarbunga plumes towards the station of Hohenpeissenberg is provided using the FLEXPART Lagrangian particle dispersion model offline coupled with the WRF_ARW atmospheric model. The simulation period is 18-26 September 2014. We assume a constant $SO_2$ release rate of 119 kilotons per day as reported by Gíslason et al. (2015) from near the source $SO_2$ measurements during the first weeks of the eruption. Similar emission rates are also suggested by Schmidt et al. (2015) through comparisons between NAME simulations (UK Met Office's Numerical Atmospheric-dispersion Modelling Environment) and OMI satellite retrievals. The emission height is set between 0 and 3500 m above ground level, consistent throughout the simulation period. The establishment of an anticyclonic flow over the British Isles on 21 September 2014 (not shown here) resulted in the separation of the volcanic $SO_2$ field into two distinct plumes (Figure 2a). On 22 September the primary plume (plume 1) becomes stagnant over the topographic barrier of the Alps (Figure 2b). The secondary plume is advected southwards by the intense northerly winds over the North Sea. The two plumes overlap at about 09:00-11:00 UTC. Taking a closer look at the surface $SO_2$ values sampled during this event by surface air quality stations in the Netherlands, several days of enhanced $SO_2$ were discovered, which indicate an area of stagnation or blocking of the flow. Trajectory calculations performed at the Royal Netherlands Meteorological Institute (KNMI) correspond well to the calculations shown in Figure 2, but also show that the air parcels stayed over Northern Europe for some time after a very fast flow over the North Sea, which agrees with the spikes found in the surface $SO_2$ records observed over the Netherlands during a period of several days.

The high $SO_2$ concentrations, which were recorded almost simultaneously at stations over Europe in various sites during the period 21-29 September 2014, are thus associated with the activity of Bárðarbunga volcano (Ialongo et. al., 2015; Table A1 see Appendix A). This is also supported by the back trajectories analysis performed with the HYSPLIT dispersion model that is shown in Figure 3. All back trajectories start at 12:00 UTC on the day of maximum $SO_2$ observations for each one of the Brewer stations and indicate that the arrival of air masses originated from Iceland.

As shown in Figure 4a, the $SO_2$ plume was detected by the Brewer instruments located in the passage of the volcanic $SO_2$ plume and from different ground based networks. However, no co-incident measurements were available from the OMI and GOME-2 overpasses at the time of the high $SO_2$ excursions. Also it should be noted here that no enhanced $SO_2$ columns were detected by the Brewers located outside of the geographical area covered by the volcanic plume (Fig. 4b). In all volcanic cases we have applied a criterion according to which each daily average from either OMI or GOME-2 should be calculated if and only if more than half of the individual overpasses had data at a given day.

The eruption took place at the beginning of September 2014 and several European countries experienced high concentrations of $SO_2$ at ground level during the rest of September. Figure 5 shows the response of ground-level AirBase stations under the plume located within 150 km from the nearest Brewer station plotted together with the co-incident Brewer $SO_2$ column measurements. Interestingly, it suggests that the highest amount of the $SO_2$

column measured by the majority of the Brewers during 21 September 2014 due to the volcano reached the surface with a time lag of about one day. The high volcanic concentrations were successfully measured by the ground-based Airbase network. Due to strong European efforts over the last decades to reduce $SO_2$ emissions, high concentrations of $SO_2$ are now quite rare in Western Europe (e.g. Vestreng et al., 2007) except in the areas affected by industrial or shipping emissions. In-situ air quality stations observed high values of $SO_2$ at the ground level in the coast of France, in the United Kingdom, the Netherlands and Germany between 21 and 25 September 2014. This all points towards a volcanic episode with a large spatial extent.

As can be seen from Figure 4a, the highest $SO_2$ column departures from the pre-volcanic baseline were observed from 21 to 22 September 2014. The mean $SO_2$ column measured by the Brewers under the plume was $2.4 \pm 0.8$ DU, which was five times greater than the mean column of $SO_2$ measured by the Brewers outside of the plume (-$0.1 \pm 0.1$ DU) by 2.5 DU on average. The "error bars" show the standard deviation of the daily $SO_2$ values of all stations during the non-perturbed 10 day period prior to the volcanic eruption. These differences provide rough estimates of the additional $SO_2$ loading induced by the volcanic eruption over Europe which exceeds 3σ. Comparison between satellite data and Brewer are limited for interpretation because satellite measurements are sparse, represent an average $SO_2$ column over a relatively large satellite pixel, while the Brewer observations are designed to provide a local point measurement. In spite of the sparsity of OMI observations post Bárðarbunga volcanic eruption, satellite data were used for assimilation in the $SO_2$ analyses and forecasts produced with the MACC (Monitoring Atmospheric Composition and Climate) system (http://atmosphere.copernicus.eu/). This near-real-time forecasting system assimilates satellite observations to constrain modelling forecasts (Inness et al., 2015; Flemming et al., 2015). The OMI instrument aboard the AURA satellite provided information about concentrations of volcanic $SO_2$ emitted by the Icelandic Bárðarbunga volcano on 20 September; these observations were assimilated in 2014 by the MACC system in cases of volcanic eruptions, i.e. when OMI values exceeded 5 DU. As shown in Figure 6 (the charts of total column $SO_2$ are taken from the website http://atmosphere.copernicus.eu/) the subsequent forecasts capture the transport of the plume of volcanic $SO_2$ southward, while spreading over the continent on 21 and 22 September. The plume stretched all the way from Finland through Poland, Germany and France, to southern England. A parallel forecast, for which no OMI data were used (Fig. 6, right), did not show any elevated $SO_2$ values, confirming that 'normal' emissions of $SO_2$ (including shipping and industrial activities) could not explain the observed situation.

Finally, it should be mentioned here that the thin aerosol layer that has been detected by the PollyXT lidar (Engelmann et al., 2015) over Leipzig at around 2-3 km on 23 and 24 of September 2014 was mostly associated with volcanic ash advection (Figure 7). A corresponding cluster analysis of all 155 hourly HYSPLIT back trajectories during this period and for the heights of the layer detected by the lidar (~2.5-3.5km) is shown in Figure 8. The increased wind shear that is evident between these heights does not allow a robust characterization of the air masses. However, the source contribution of about 20% from Icelandic air masses supports the volcanic origin of the detected plume. During volcanic eruptions, ash and $SO_2$ may be injected to different altitudes and may follow different trajectories for long-range transport. EARLINET lidars can provide alerts on volcanic ash dispersion over Europe, especially when the systems are employed with depolarization capabilities (e.g. Pappalardo et al., 2013). For the Brewer network capabilities and the Hohenpeissenberg station, Figures 7

and 8 demonstrate that similar approach can be applied to contribute towards an early warning synergistic tool, as evidenced in the Bárðarbunga case study. The role of the Brewer stations in this system will be the early detection of $SO_2$ plumes transported over continental areas that would trigger the associated forecasting systems (models and networks).

## 3.2 The 2011 Nabro Volcano plume

A major eruption of Mt. Nabro, a 2218 m high volcano on the border between Eritrea and Ethiopia (13.37$^{o}$ N, 41.7$^{o}$ E), occurred on 12–13 June, 2011. The volcanic eruption injected ash, water vapour and an estimated 1.3–2.0 Tg of $SO_2$ into the upper troposphere and lower stratosphere (Fairlie et al., 2014 and references therein). In the first phase of the eruption, the main transport pattern of emitted $SO_2$ followed the strong anticyclonic circulation over the Middle East and Asia associated with the Asian summer monsoon at that time of year (Clarisse et al., 2014 and references therein). In the first month after the eruption stratospheric aerosols were mainly observed over Asia and the Middle East, and by day 60 covered the whole Northern Hemisphere. Estimated aerosol altitudes from various instruments were between 12 and 21 km (Clarisse et al., 2014). By July 2011 Nabro had cumulatively emitted 5 to 10 percent of what was released by Mount Pinatubo in 1991 (~20 Tg) ranking it among the largest $SO_2$ emissions in the tropical stratosphere (up to at least 19 km) since Pinatubo (Krotkov et al., 2011). Sulfur dioxide signals of volcanic origin were detected both by Brewer and satellite measurements over East Asia where the volcanic $SO_2$ plume was transported, as demonstrated in Figure 9 and 10a. Measurements were taken by Brewer in Taipei, Taiwan, Asia. This is also evident from the back trajectories analysis performed with the HYSPLIT dispersion model for Taipei (Taiwan) as shown in Figure 10a. The analysis indicates that the upper tropospheric air masses arriving at Taipei on June 19, when the peak in $SO_2$ is observed, originate from Africa.

The Nabro volcanic plume was mainly transported to the East Asia and was detected by various satellite instruments which provide better spatial coverage than the Brewers. A special case study focuses on discrepancies found between ground based and satellite observations of the volcanic $SO_2$ plume. Brewer located in Tenerife, Spain detected an increase in the $SO_2$ column, which was not clearly detected by the OMI and GOME-2 satellite overpasses.

More specifically, Figure 10b shows back trajectories from Izaña (Tenerife) during 19-29 June 2011 at 15, 17.5 and 20 km heights. It appears that the upper tropospheric-lower stratospheric air masses arriving at Tenerife during 19-29 June originated from Nabro. In June 2011 the Nabro volcano ash plume was detected by the Micropulse Lidar (MPL) located at Santa Cruz de Tenerife (The Canary Islands, Spain). The volcanic plume height ranged from 12 km on 19 June to 21 km on 29 June (Sawamura et al., 2012). Figure 11 shows the columnar $SO_2$ departures from the unperturbed 10 day pre-volcanic baseline measured by the Brewer at Izaña following Nabro. The daily mean $SO_2$ departures (Figure 11) show a 0.3 DU increase at the beginning of the event (19 June), reaching 0.6 DU on 29 June when the layer is found at higher altitude. The signal is not strong and is near the error of 0.5 DU estimated for $SO_2$ measurement (Stanek, personal communication) but the observations are consistent (independent of the ozone and air mass), since at Izaña about 100 $O_3$/$SO_2$ measurements per day are performed resulting to reduced standard errors associated with daily means as

compared to individual observations. The Langley calibration is tracked between calibrations by measurements of the internal lamp (Langley and lamp are shown in Supplement Figure S1). The increase in $SO_2$ due to the passage of the Nabro volcano plume over the Canary Islands is significant using both methods (Figure 11).

In this case the Brewer at Izaña has been able to detect an $SO_2$ plume at high altitude from a volcano located 7,000 km from the Canary Islands, indicating that the Brewer network is sensitive enough to be incorporated in columnar $SO_2$ monitoring from volcanic eruptions in worldwide networks.

The case of the 2011 Nabro eruption shows an example of the importance of the Brewer spectrophotometers in measuring and detecting changes in $SO_2$ amounts in the atmosphere due to volcanic eruptions, in cases where signal in the satellite overpasses is low. This is true for the case of Izaña (Tenerife) where it appears that OMI and GOME-2 did not clearly detect increases in $SO_2$ column of volcanic origin between 19 June and 1 July as it was the case with the Brewer instrument (Figure 11). During some days between 19 June and 1 July, the Brewer $SO_2$ columns at Izaña rose above the uncertainty of 0.5 DU for the Brewer $SO_2$ measurements at this station, whereas the satellite $SO_2$ columns stayed mostly within the uncertainty of 0.4 DU estimated for OMI and GOME-2 satellite retrievals.

These findings can provide clues on the detection limits of such events from a well calibrated Brewer network and a space-borne instrument. They need further clarification with more Brewers and a larger number of cases.

**3.3 The case of the 2010 Eyjafjallajökull volcanic eruption**

The Eyjafjallajökull volcano, Iceland (63.63° N, 19.6215° W; 1666 m asl) erupted explosively on 14 April 2010 and continued to emit ash and gas until 24 May (Flentje et al., 2010; Thomas and Prata, 2011; Stohl et al., 2011; Flemming and Inness, 2013). Despite the relatively modest size of the eruption, the prevailing wind conditions advected the volcanic plume toward the south-east leading to unprecedented disruption to air traffic in Western Europe. This caused significant financial losses for the airlines and highlights the importance of efficient volcanic cloud monitoring and forecasting. Results demonstrate that the eruption can be divided into an initial ash rich phase (14-18 April), a lower intensity middle phase (19 April until early May) and a final phase (4-24 May) where considerably great quantities both ash and $SO_2$ were released (Thomas and Prata, 2011).

Figure 12 shows the responses of Brewer stations under the volcanic $SO_2$ plume and the average of Brewer stations outside of the plume together with OMI and GOME-2 satellite observations. We determined 10 stations being under the plume in 2010 and 10 stations being outside of the plume based on the analysis of forward and backward trajectories of air masses following the volcanic eruption. The stations determined to be under the plume in 2010 (shown in Figure 12a) are Sodankylä, Obninsk, Manchester, De Bilt, Uccle, Belsk, Reading, Hohenpeissenberg, Davos and Arosa. The stations determined to be outside of the plume are Vindeln, Oslo, Norrkoeping, Copenhagen, Hradec Kralove, Aosta, Kislovodsk, Rome, Thessaloniki and Athens (Figure 12b).

We should note here that volcanic clouds can be rather narrow plumes with diameters on the order of a few tens kilometers (e.g. Stohl et al., 2011; Webley et al., 2012; Thorsteinsson et al., 2012; Kristiansen et al., 2012;

Kokkalis et al., 2013) and thus it is possible that a detected volcanic layer at a specific station is not observed by neighbouring stations. The measurements at Uccle and De Bilt that are located at a horizontal distance of 150 km are different during the Eyjafjallajökull episode and provide a formidable example. On 2 May 2010 the mean daily $SO_2$ is 5.8 DU at De Bilt and -0.2 DU at Uccle. As seen at the corresponding back trajectories for that day in Figure 13, air masses originating from Iceland arrive at De Bilt at heights 6-7 km and have been probably transporting the volcanic cloud over that station. In contrast similar back trajectories on the same day for the case of Uccle indicate transport of air masses from Iceland but at lower heights (3-4 km) that were probably not affected by the volcanic emissions. In another case of transport on 11 May 2010, Uccle was outside of the plume (see Figure 13) while the mean daily $SO_2$ for De Bilt was 0.9 DU. On 18 May 2010, both De Bilt and Uccle stations detected a volcanic cloud with $SO_2$ daily means of 1.7 DU and 1.2 DU respectively. To summarize, in spite the proximity of Uccle and De Bilt the transport heights and trajectories can have a different result in transporting volcanic gases.

In Table A1 of Appendix A, we present the dates when the examined Brewer stations were either under or outside of the volcanic $SO_2$ plume. Careful analysis of the trajectories of the volcanic plumes in 2010 and 2014 helped verify these analyses. The distinction between stations outside of the plume and stations under the plume was done as follows: whenever $SO_2$ at each station measuring exceeded 2 DU ($2\sigma$) back trajectories were calculated and the origin was compared to the location of the volcanic eruption. All these stations have been considered to be under the $SO_2$ plume. All other stations, for which columnar $SO_2$ amounts were within $2\sigma$ and were not originating from the area of the eruption, were considered to be outside of the volcanic $SO_2$ plume.

As we can see from Figure 12, the columnar $SO_2$ departures at stations located under the passage of the volcanic $SO_2$ plume exceeded 0.3 DU (reaching 1.5 DU in some cases) whereas at stations located outside of the plume, the columnar $SO_2$ departures did not exceed 0.3 DU. Moreover, during the explosive phase 2 there were three main periods in which the volcanic aerosol content was observed by EARLINET over Europe: 15-26 April, 5-13 May and 17-20 May. These periods were determined from measurements of the integrated backscatter at 532 nm in the volcanic layers (Pappalardo et al., 2013). We estimate high $SO_2$ columnar departures measured by the Brewers under the plume during 14 April and 23 May 2010 up to 6.0 DU (e.g. Arosa, 18 May 2016).

We note here that the ash cloud caused further disruptions to air transportation on 4-5 May and 16-17 May 2010, particularly over Ireland and the UK. The average $SO_2$ columnar departures measured by the Brewers under the plume in the UK (Manchester and Reading) during these two periods were estimated to $1.1 \pm 0.3$ DU and $1.5 \pm 0.4$ DU respectively. These amounts were higher than the amounts measured outside of the plume (-0.1 ± 0.2 DU and -0.1 ± 0.1 DU, accordingly) almost by 1.4 DU on average.

**3.4 An eruption of larger scale importance – The 2008 Kasatochi case**

The eruption of Kasatochi volcano on 7-8 August 2008 injected large amounts of material and $SO_2$ into the troposphere and lower stratosphere of the northern middle latitudes during a period of low stratospheric aerosol background concentrations. The Kasatochi volcano in the central Aleutian Islands of Alaska (52.17$^o$ N, 175.51$^o$ W) erupted three times between 2201 UTC on 7 August and 0435 UTC on 8 August 2008 (Bitar et al., 2010).

Aerosols from the volcanic eruption were detected by lidar in Halifax shortly after the eruption (Bitar et al., 2010). The total mass of $SO_2$ injected into the atmosphere by the eruption is estimated at 1.7 Tg, with about 1 Tg reaching the stratosphere (above 10 km asl) (Kristiansen et al., 2010).

We have studied the columnar $SO_2$ amounts following the Kasatochi eruption in August 2008 from ground based and satellite data. Figure 14 shows the columnar $SO_2$ departures from the unperturbed 10 day pre-volcanic period over Canada/USA and Europe during the bimonthly period August-September 2008 as measured by the Brewers in comparison with the satellite observations by OMI and GOME-2.

The $SO_2$ plume was clearly seen by the Brewers in Canada/USA (Figure 14) and it was also detected by the majority of the Brewers in Europe with a delay by about 3 days. The total $SO_2$ columnar departures averaged over Canada during the period 12-20 August 2008 are estimated to 0.9 ± 0.2 DU. Accordingly over Europe, we estimate a mean $SO_2$ columnar departure of 1.0 ± 0.1 DU during the period 15-22 August 2008. This number gives a rough estimate of the average volcanic $SO_2$ column measured by the Brewers over Europe. We note here

that the e-folding time of the Kasatochi $SO_2$, i.e. the time where the volcanic $SO_2$ amount decayed, was estimated to be about 8-9 days (Krotkov et al., 2010).

The high amounts of $SO_2$ and the variability of $SO_2$ measured in Europe by the Brewers after the eruption of Kasatochi in August 2008 are in line with OMI and GOME-2 satellite observations. More specifically, OMI

(TRM) shows an average $SO_2$ columnar departure of 0.5 ± 0.1 DU during the period 15-22 August 2008, OMI (STL) an average $SO_2$ columnar departure of 0.4 ± 0.1 DU and GOME-2 an average $SO_2$ columnar departure of 0.8 ± 0.1 DU respectively.

The Brewer data have been correlated with those from OMI and GOME-2. The Pearson's correlation coefficients between the three datasets were all highly statistically significant (>99%). The correlation between $SO_2$ from the

Brewers and $SO_2$ from GOME-2 at 19 stations averaged over Europe is +0.86 (t-value = 12.54, p-value < 0.0001, N = 59). Accordingly, the correlation between Brewer and OMI (TRM) $SO_2$ data is +0.86 (t-value = 11.77, p < 0.0001, N = 50) and between GOME-2 and OMI (TRM) data is +0.92 (t-value = 16.32, p < 0.0001, N = 48). Same correlations are found for the Brewer-OMI (STL) and GOME-OMI (STL) data pairs. These correlations

were calculated from 60 daily averages during the Kasatochi volcanic eruption in August-September 2008. The statistical tests gave significant results and verified the capability of the Brewers in detecting natural $SO_2$ emitted by volcanoes when the volcanic plume of $SO_2$ passes over the ground sites. We note here that there is a general consistency between all three datasets (Brewers, OMI and GOME-2) on the changes in $SO_2$ column following the Kasatochi volcanic eruption.

Table 5 summarises the correlation coefficients between the mean columnar $SO_2$ measured by all Brewers over Europe and provided by the satellite products of OMI and GOME-2 during the globally extended Kasatochi event. The correlation coefficients have high statistical significance explaining more than 70% of the total variance between the columnar $SO_2$ measurements from ground and space in the case of Kasatochi. However,

the discrepancies found between satellite and Brewer observations during the other volcanic eruptions could be

impacted by sparsity of coincident measurements, and thus cannot confirm or deny Kasatochi case findings at high significance levels.

**4 Conclusions**

In this work we provide evidence that the current network of Brewer spectroradiometers is capable of identifying columnar $SO_2$ plumes of volcanic origin. The study is based on the results from the three largest volcanic eruptions (VEI $\geq$4) in the past decade when elevated $SO_2$ plumes have passed over Brewer stations in the Northern Hemisphere. The analysis included a fourth eruption, namely Bárðarbunga, because it has perturbed the $SO_2$ regime over large parts of Europe and extended from the free troposphere down to the surface. Back and forward trajectory analysis have been used to aid in identifying and selecting measurements taken under and outside of the volcanic $SO_2$ plume. When the plume was overpassing a site, the $SO_2$ signal was found to be quite high, exceeding 3σ of daily values relative to the average levels taken during the unperturbed measurements over ten days preceding each eruption. On the average the mean $SO_2$ columnar amount to be attributed to the volcano is estimated to be on the order of 2 DU as discussed in section 3. In addition to the Brewer network, comparisons were made with other instruments (e.g. surface $SO_2$ sensors) that were located under the volcanic $SO_2$ plumes. Moreover, satellite measurements of columnar $SO_2$ from OMI and GOME-2 collocated with the Brewer network were used for comparisons.

From the results discussed in section 3 some general remarks can be put forward concerning $SO_2$ levels and detection time after the eruption. Starting with the Kasatochi eruption, as it appears from Figure 14, the plume can be detected 4 days after the eruption over Canada and the US and about 7 days over Europe with an average amplitude on the order of 2 DU compared to the unperturbed ten day pre-volcanic period (baseline). All estimates are based obviously on measurements taken under the plume. The Kasatochi eruption provided a formidable example for a volcanic $SO_2$ plume to be observed not only by the ground based instruments, but from space-borne as well (OMI and GOME-2). Relative to the undisturbed period before Kasatochi the amplitude of the signal is 2 DU for GOME-2 and 1.5 DU for OMI. The results for the other volcanic eruptions are similar for the Brewer network, but unfortunately because of the sparsity of satellite overpassing the Brewer stations the satellite data concur with those from the Brewers only in Kasatochi. Based on the above discussion it appears that currently no single network can independently and fully monitor the evolution of volcanic $SO_2$ plumes. Among a few reasons are lack of measurements during peak values, complications from meteorological events, ejection heights and exposure conditions. The evidence presented here points that combination of observations from various instruments, aided by chemical transport models and operated in synergy could address such a complex issue.

The combination of the above discussed observation and modelling tools can assist in detecting existing volcanic plumes, but also in forecasting their evolution, which can have importance not only to the air traffic warning but also to air pollution in the lower layers of the atmosphere. Therefore, an automated source receptor modelling tool could be proposed as follows: a modelling system based on FLEXPART and HYSPLIT backward-trajectory simulations could be automatically triggered whenever high $SO_2$ values are detected at a Brewer station above a

specific threshold (e.g. 3σ of station's daily values) or when a lidar instrument detects highly depolarizing layers that were not advected from a geographical location over a desert. The operational use of such a synergistic activity could provide near-real-time and forecasting information on the evolution of volcanic episodes and also develop a comprehensive database of measurements useful to improve model forecasts. This new well-tuned and organised synergistic activity of monitoring networks, observations and modelling from ground and space could create a challenging monitoring tool for volcanic and other extreme emissions, which form the basis towards a new regional $SO_2$ columnar forecasting facility.

## 5 Data availability

$SO_2$ columns at Churchill, Goose, Edmonton, Regina, Saturna Island and Toronto in Canada, Taipei in Taiwan, Boulder and Mauna Loa in the US were obtained from the World Ozone and Ultraviolet Radiation Data Centre (WOUDC; http://www.woudc.org/, last access: 10 October 2016) and the NOAA-EPA Brewer Spectrophotometer UV and Ozone Network (NEUBrew; http://www.esrl.noaa.gov/gmd/grad/neubrew/, last access: 10 October 2016). OMI and GOME-2 satellite $SO_2$ data products were downloaded from the Aura Validation Data Center (AVDC) (available from: http://avdc.gsfc.nasa.gov/index.php?site=245276100, last access: 10 October 2016). Surface $SO_2$ concentrations over Europe were acquired from the European Environment Agency databases (AirBase) (http://www.eea.europa.eu/data-and-maps/data/aqereporting-1#tab-european-data, last access: 10 October 2016).

## Appendix A

**Table A1. Dates in which the Brewers were determined to be under or outside of the volcanic $SO_2$ plume, based on analysis of back trajectories of the volcanic plumes in 2010 and 2014. The distinction between stations outside of the plume and stations under the plume was done as follows: At each station measuring $SO_2$ exceeding 2 DU (2σ) we calculated back trajectories and found that their origin was at the volcanic eruption. All these stations have been considered to be under the $SO_2$ plume. All other stations, for which columnar $SO_2$ amounts were within 2σ and were not originating from the area of the eruption, were considered to be outside of the volcanic $SO_2$ plume. During the Kasatochi eruption all Brewers were considered to be under the volcanic $SO_2$ plume.**

| Station | LAT (deg) | LON (deg) | ALT (m) | 2010 | 2014 |
|---|---|---|---|---|---|
| Sodankylä | 67.36 | 26.63 | 180 | 20/4 | 27/9 and 29/9 |
| Vindeln | 64.24 | 19.77 | 225 | Outside the plume | 29/9 |
| Jokioinen | 60.82 | 23.50 | 106 | No data | 27/9 |
| Oslo | 59.90 | 10.73 | 50 | Outside the plume | Outside the plume |
| Norrkoeping | 58.58 | 16.15 | 43 | Outside the plume | 30/9 |
| Copenhagen | 55.63 | 12.67 | 50 | Outside the plume | 24/9 |
| Obninsk | 55.10 | 36.60 | 100 | 23/4 and 25/4 | 28/9 |
| Manchester | 53.47 | -2.23 | 76 | 16/5 | 21/9 |
| Warsaw | 52.17 | 20.97 | 107 | No data | Outside the plume |
| De Bilt | 52.10 | 5.18 | 24 | 2/5, 11/5, 18/5 | 21/9 |
| Belsk | 51.84 | 20.79 | 180 | 10/5 | Outside the plume |
| Reading | 51.44 | -0.94 | 66 | 16/5 | 21/9 |
| Uccle | 50.80 | 4.36 | 100 | 18/5 | 21-22/9 |
| Hradec Kralove | 50.18 | 15.84 | 285 | Outside the plume | 24/9 |
| Hohenpeissenberg | 47.80 | 11.01 | 985 | 18/5 | 22/9 |
| Davos | 46.81 | 9.84 | 1590 | 27/4 and 18-19/5 | Outside the plume |
| Arosa | 46.78 | 9.67 | 1840 | 18/5 | Outside the plume |
| Aosta | 45.74 | 7.36 | 569 | Outside the plume | 23/9 |
| Kislovodsk | 43.73 | 42.66 | 2070 | Outside the plume | Outside the plume |
| Rome | 41.90 | 12.52 | 75 | Outside the plume | Outside the plume |

| Thessaloniki | 40.63 | 22.95 | 60 | Outside the plume | No data |
| Athens | 37.99 | 23.78 | 191 | Outside the plume | Outside the plume |

## Acknowledgements

The authors would like to particularly thank Andreas Engel and two anonymous reviewers for their valuable comments. This research was supported by the Copernicus Atmosphere Monitoring Service (CAMS), the Mariolopoulos-Kanaginis Foundation for the Environmental Sciences and the project of EUMETSAT, O3M

SAF. We acknowledge the COST Action ES1207 "A European Brewer Network (EUBREWNET)", the WMO World Ozone and Ultraviolet Radiation Data Centre (WOUDC), the NOAA-EPA Brewer Spectrophotometer UV and Ozone Network (NEUBrew), the NASA GSFC Aura Validation Data Center (AVDC) and the EEA European air quality database (AirBase).

This project has received funding from the European Union's Horizon 2020 research and innovation programme

under grant agreement No 654109.

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

**Table 1. Volcanic eruptions in the past decade considered in this study.**

| Volcano | Latitude | Longitude | Elevation (asl) | Period of Eruption | VEI* |
|---|---|---|---|---|---|
| Okmok, Alaska | 53.43°N | 168.13°W | 1073 m | 12 July - 19 August 2008 | 4 |
| Kasatochi, Alaska | 52.17°N | 175.51°W | 300 m | 7-8 August 2008 | 4 |
| Sarychev, Russia | 48.1°N | 153.2°E | 1496 m | 12-17 June 2009 | 4 |
| Eyjafjallajökull, Iceland | 63.63°N | 19.62°W | 1666 m | 14 April - 23 May 2010 | 4 |
| Grímsvötn, Iceland | 64.42°N | 17.33°W | 1725 m | 21-25 May 2011 | 4 |
| Nabro, Africa | 13.37°N | 41.70°E | 2218 m | 12-13 June 2011 | 4 |
| Tolbachik, Russia | 55.83°N | 160.33°E | 3611 m | 27 November 2012 - 22 August 2013 | 4 |
| Bárðarbunga, Iceland | 64.64°N | 17.56°W | 2005 m | 31 August 2014 - 28 February 2015 | 0 |

*taken from the Smithsonian Institution Global Volcanism Program

**Table 2. Stations with accessible SO$_2$ column data from Brewers analysed in this study. Stations are sorted from high to lower northern latitudes.**

| | Station | Latitude | Longitude | Elevation asl (m) | Instruments | Data source |
|---|---|---|---|---|---|---|
| 1 | Sodankylä | 67.36 | 26.63 | 180 | Brewer MKII 037 | FMI |
| 2 | Vindeln | 64.24 | 19.77 | 225 | Brewer MKII 006 | SMHI |
| 3 | Jokioinen | 60.82 | 23.50 | 106 | Brewer MKIII 107 | FMI |
| 4 | Oslo | 59.90 | 10.73 | 50 | Brewer MKV 042 | U_Oslo |
| 5 | Churchill | 58.74 | -93.82 | 16 | Brewer MKII 026, Brewer MKIV 032, Brewer MKIII 203 | WOUDC |
| 6 | Norrkoeping | 58.58 | 16.15 | 43 | Brewer MKIII 128 | SMHI |
| 7 | Copenhagen | 55.63 | 12.67 | 50 | Brewer MKIVe 082 | DMI |
| 8 | Obninsk | 55.10 | 36.60 | 100 | Brewer MKII 044 | IEM-SPA |
| 9 | Edmonton | 53.55 | -114.10 | 766 | Brewer MKII 055, Brewer MKIV 022 | WOUDC |
| 10 | Manchester | 53.47 | -2.23 | 76 | Brewer MKIII 172 | U_Manchester |
| 11 | Goose Bay | 53.29 | -60.39 | 39 | Brewer MKII 018 | WOUDC |
| 12 | Warsaw | 52.17 | 20.97 | 107 | Brewer MKIII 207 | PAS-IGF |
| 13 | De Bilt | 52.10 | 5.18 | 24 | Brewer MKIII 189 | KNMI |
| 14 | Belsk | 51.84 | 20.79 | 180 | Brewer MKII 064 | PAS-IGF |
| 15 | Reading | 51.44 | -0.94 | 66 | Brewer MKIV 075, Brewer MKII 126 | U_Manchester |
| 16 | Uccle | 50.80 | 4.36 | 100 | Brewer MKII 016, Brewer MKIII 178 | RMIB |
| 17 | Regina | 50.20 | -104.71 | 580 | Brewer MKIII 111 | WOUDC |
| 18 | Hradec Kralove | 50.18 | 15.84 | 285 | Brewer MKIII 184 | CHMI-HK |
| 19 | Saturna Island | 48.78 | -123.13 | 178 | Brewer MKII 012 | WOUDC |
| 20 | Hohenpeissenberg | 47.80 | 11.01 | 985 | Brewer MKII 010 | DWD-MOHp |
| 21 | Davos | 46.81 | 9.84 | 1590 | Brewer MKIII 163 | PMOD/WRC |
| 22 | Arosa | 46.78 | 9.67 | 1840 | Brewer MKII 040, Brewer MKIII 156 | MeteoSwiss |
| 23 | Aosta | 45.74 | 7.36 | 569 | Brewer MKIV 066 | ARPA-VDA |
| 24 | Toronto | 43.78 | -79.47 | 198 | Brewer MKII 015 | WOUDC |
| 25 | Kislovodsk | 43.73 | 42.66 | 2070 | Brewer MKII 043 | RAS-IAP |
| 26 | Rome | 41.90 | 12.52 | 75 | Brewer MKIV 067 | U_Rome |
| 27 | Thessaloniki | 40.63 | 22.95 | 60 | Brewer MKII 005 | AUTH |
| 28 | Boulder | 40.03 | -105.53 | 2891 | Brewer MKIV 146 | NEUBrew |
| 29 | Athens | 37.99 | 23.78 | 191 | Brewer MKIV 001 | BRFAA |
| 30 | Izaña | 28.31 | -16.50 | 2373 | Brewer MKIII 157 | AEMET |
| 31 | Taipei | 25.04 | 121.51 | 5 | Brewer MKIII 129 | WOUDC |
| 32 | Mauna Loa | 19.54 | -155.60 | 3397 | Brewer MKIII 119 | WOUDC |

**Table 3. Rural AirBase stations analysed in this study (see text).**

| Station ID | Station name | Latitude | Longitude | Closest Brewer (within 150 km) |
|---|---|---|---|---|
| GB0583A | Middlesbrouth | 54.569 | -1.221 | Manchester |
| NL00444 | De Zilk-Vogelaarsdreef | 52.298 | 4.51 | De Bilt |
| PL0105A | Parzniewice | 51.291 | 19.517 | Belsk |
| NL00133 | Wijnandsrade-Opfergeltstraat | 50.903 | 5.882 | Uccle |
| GB0038R | Lullington Heath | 50.794 | 0.181 | Reading |
| CH0005R | Rigi | 47.067 | 8.463 | Arosa |
| CH0002R | Payerne | 46.813 | 6.944 | Aosta |

5 **Table 4. SO₂ column departures at mid-latitude stations averaged in bimonthly periods following volcanic eruptions.**

| (a) | Latitude | August-September 2008 (Kasatochi) | | April-May 2010 (Eyjafjallajökull) | | September-October 2014 (Bárðarbunga) | |
|---|---|---|---|---|---|---|---|
| | | mean | σ | mean | σ | mean | σ |
| Sodankylä | 67.36 | 0.6 | 2.1 | 0.1 | 0.7 | -0.5 | 1.8 |
| Vindeln | 64.24 | 0.4 | 1.4 | 0.0 | 0.4 | -0.2 | 0.9 |
| Jokioinen | 60.82 | 0.5 | 0.6 | * | * | 0.4 | 0.5 |
| Oslo | 59.90 | * | * | 0.7 | 0.6 | -0.1 | 1.0 |
| Churchill | 58.74 | 0.6 | 0.8 | -0.3 | 1.1 | 0.4 | 1.0 |
| Norrkoeping | 58.58 | 0.4 | 0.8 | -0.1 | 0.2 | 0.1 | 0.8 |
| Copenhagen | 55.63 | 0.3 | 0.8 | 0.5 | 0.9 | -0.4 | 0.7 |
| Obninsk | 55.10 | * | * | 0.1 | 0.5 | 0.3 | 0.9 |
| Edmonton | 53.55 | 0.4 | 0.6 | 0.4 | 0.4 | 0.0 | 0.4 |
| Manchester | 53.47 | 0.6 | 0.7 | 0.0 | 0.6 | 0.4 | 1.6 |
| Goose Bay | 53.29 | 0.2 | 0.4 | * | * | 0.3 | 0.3 |
| Warsaw | 52.17 | * | * | * | * | 0.1 | 0.4 |
| De Bilt | 52.10 | 0.1 | 0.9 | -0.3 | 0.9 | 0.2 | 0.8 |
| Belsk | 51.84 | 0.3 | 0.6 | -0.4 | 0.4 | 0.4 | 0.5 |
| Reading | 51.44 | 0.2 | 0.7 | 1.2 | 1.2 | 0.3 | 1.7 |
| Uccle | 50.80 | 0.1 | 0.6 | -0.5 | 0.6 | 0.7 | 1.3 |
| Regina | 50.20 | 0.0 | 0.9 | * | * | * | * |
| Hradec Kralove | 50.18 | 0.2 | 0.4 | -0.3 | 0.4 | -0.6 | 0.7 |
| Saturna Island | 48.78 | 0.4 | 1.1 | 0.0 | 0.2 | 0.4 | 0.5 |
| Hohenpeissenberg | 47.80 | 0.0 | 0.5 | 0.5 | 0.6 | -0.1 | 1.6 |
| Davos | 46.81 | 0.2 | 0.5 | -0.1 | 0.3 | -0.1 | 0.2 |
| Arosa | 46.78 | 0.6 | 1.5 | -0.5 | 1.5 | -0.1 | 0.5 |
| Aosta | 45.74 | -0.1 | 0.6 | 0.0 | 0.6 | -0.6 | 0.8 |
| Toronto | 43.78 | 0.5 | 1.0 | -0.2 | 0.5 | 0.4 | 0.5 |
| Kislovodsk | 43.73 | -0.3 | 0.3 | -0.1 | 0.3 | 0.2 | 0.2 |
| Rome | 41.90 | -0.1 | 1.1 | -0.8 | 1.3 | -0.2 | 0.5 |
| Thessaloniki | 40.63 | 0.4 | 0.7 | -0.7 | 0.9 | * | * |
| Boulder | 40.03 | 0.1 | 0.5 | 0.1 | 0.9 | * | * |
| Athens | 37.99 | 0.9 | 0.8 | -0.4 | 0.6 | 0.0 | 0.4 |
| (b) | | mean ± st. error (N) | | mean ± st. error (N) | | mean ± st. error (N) | |
| All Brewers | | 0.29 ± 0.03 (1051) | | -0.04 ± 0.03 (1064) | | 0.07 ± 0.03 (861) | |
| GOME-2 | | 0.23 ± 0.02 (1057) | | 0.08 ± 0.01 (971) | | -0.03 ± 0.02 (677) | |
| OMI (TRM) | | 0.15 ± 0.02 (741) | | 0.00 ± 0.02 (438) | | 0.01 ± 0.02 (395) | |
| OMI (STL) | | 0.12 ± 0.01 (741) | | 0.00 ± 0.01 (438) | | 0.01 ± 0.02 (395) | |

(*) missing values are those possessing < 25 days of data in each bimonthly period, or no data.

**Table 5. Correlation coefficients between the mean columnar SO$_2$ measured by the brewers in Europe and provided by the satellite products of OMI and GOME-2 during the volcanic eruptions of Kasatochi (2008), Eyjafjallajökull (2011) and Bárðarbunga (2014) for stations located under the volcanic SO$_2$ plume.**

| Europe | August-September 2008 | April-May 2010 | September-October 2014 |
|---|---|---|---|
| Brewers and GOME-2 | **0.86** [59] (p<0.0001) | **0.31** [54] (p=0.02336) | **0.44** [39] (p=0.00496) |
| Brewers and OMI (TRM) | **0.86** [50] (p<0.0001) | (*) [23] | (*) [15] |
| Brewers and OMI (STL) | **0.86** [50] (p<0.0001) | (*) [23] | (*) [15] |
| GOME-2 and OMI (TRM) | **0.92** [48] (p<0.0001) | (*) [21] | (*) [15] |
| GOME-2 and OMI (STL) | **0.93** [48] (p<0.0001) | (*) [21] | (*) [15] |

Bold: all the above correlations are significant at confidence level 95% or greater (t-test).
(*): missing correlations are those possessing less than 30 days of data in each bimonthly period. In brackets: number of pairs.

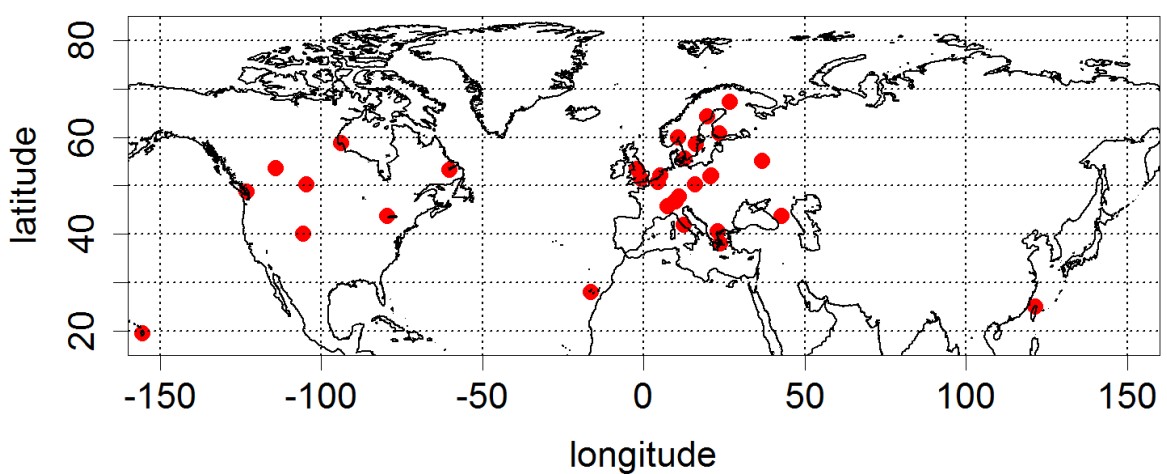

5    **Figure 1. All stations with accessible SO₂ column data from Brewers analysed in this study as listed in Table 2.**

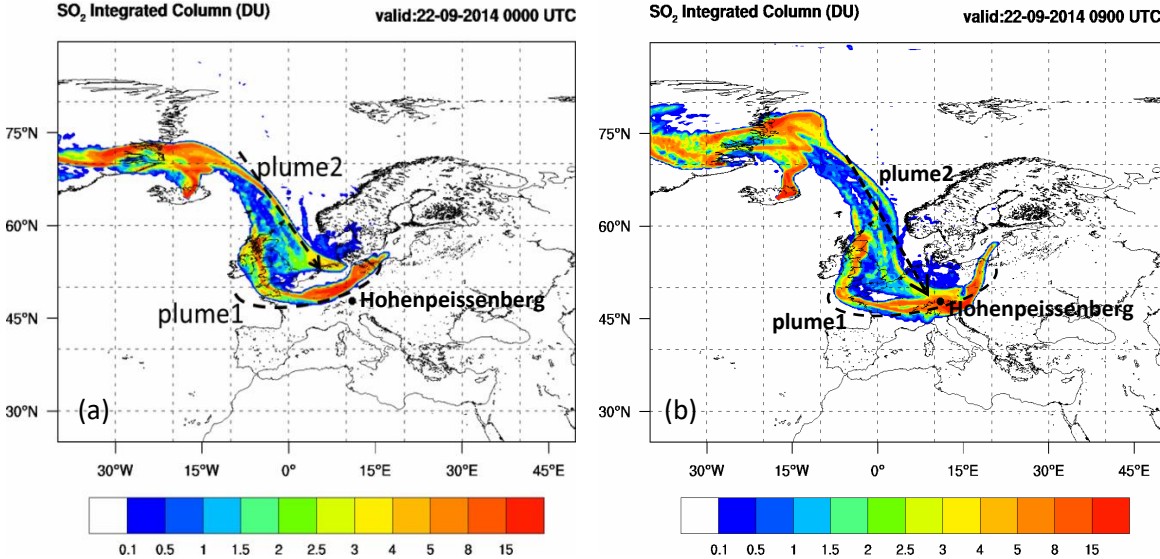

**Figure 2. Integrated column of SO₂ (DU) from Bárðarbunga emissions as simulated with FLEXPART-WRF model, a) 22 September 2014 00:00 UTC; b) 22 September 09:00 UTC. Dashed lines indicate the orientation of the two distinct plumes overlapping over central Europe.**

# Bárðarbunga 120h backward trajectories (from Brewer Stations)

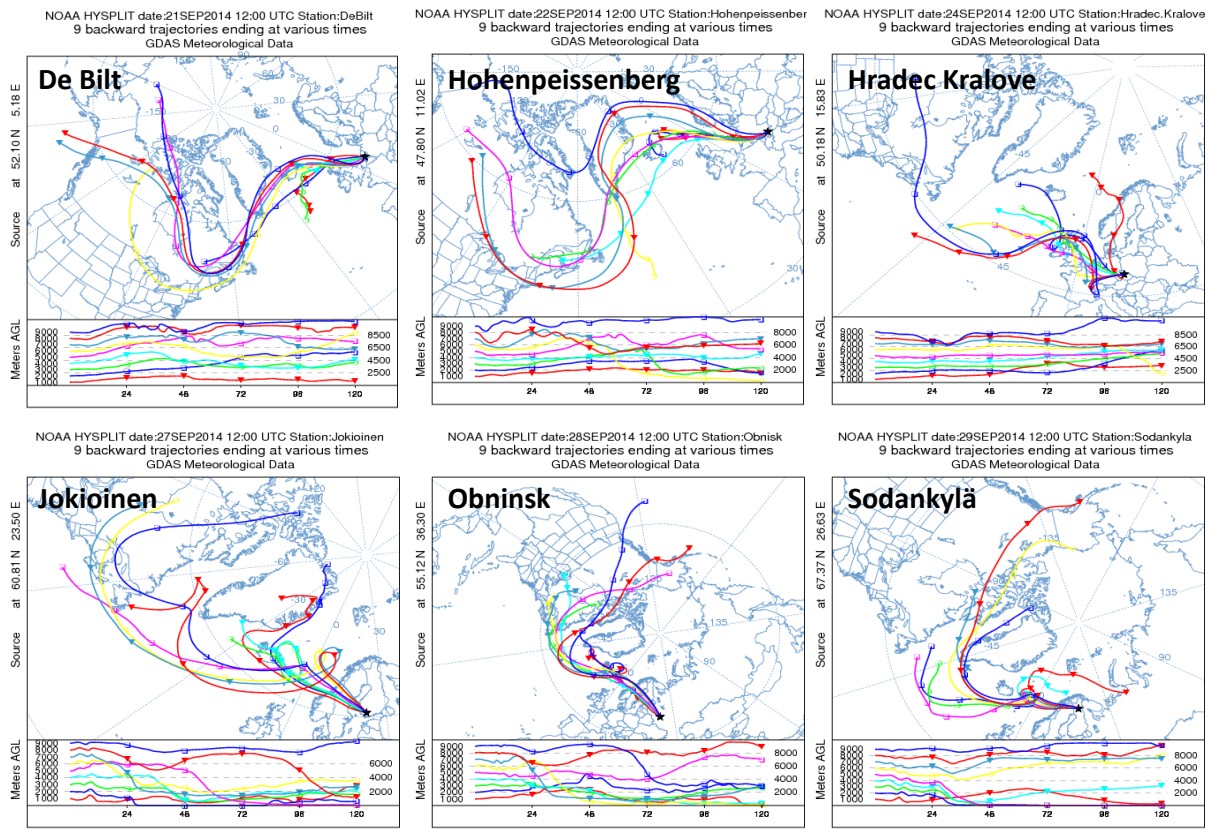

**Figure 3. HYSPLIT 120 hours back trajectories of air masses arriving on the day of maximum SO₂ records for each one of the Brewer stations at De Bilt, Hohenpeissenberg, Hradec Kralove, Jokioinen, Obninsk and Sodankylä.**

 **SO$_2$ column departures (September-October 2014)**

 **SO$_2$ column departures (September-October 2014)**

**Figure 4. Mean SO$_2$ column departures from the unperturbed 10 day pre-volcanic baseline measured by Brewers, OMI (TRM, STL) and GOME-2 during September-October 2014 over Europe following the 2014 Bárðarbunga volcanic eruption for: (a) stations under the volcanic SO$_2$ plume, and (b) stations outside of the plume. The error bars for the Brewer observations show the standard deviation of all daily values during the unperturbed 10 day period prior to the volcanic eruption. Brewer stations under the plume are: Sodankylä, Vindeln, Jokioinen, Oslo, Norrkoeping, Copenhagen, Obninsk, Manchester, De Bilt, Reading, Uccle, Hradec Kralove, Hohenpeissenberg and Aosta. Stations outside of the plume are: Warsaw, Belsk, Davos, Arosa, Kislovodsk, Rome and Athens. Each daily average from either OMI or GOME-2 was calculated if and only if more than half of the individual overpasses had data at a given day. The arrow marks the starting date of the eruption (beginning on 31 August 2014 which continued to be active throughout the whole bimonthly period and beyond).**

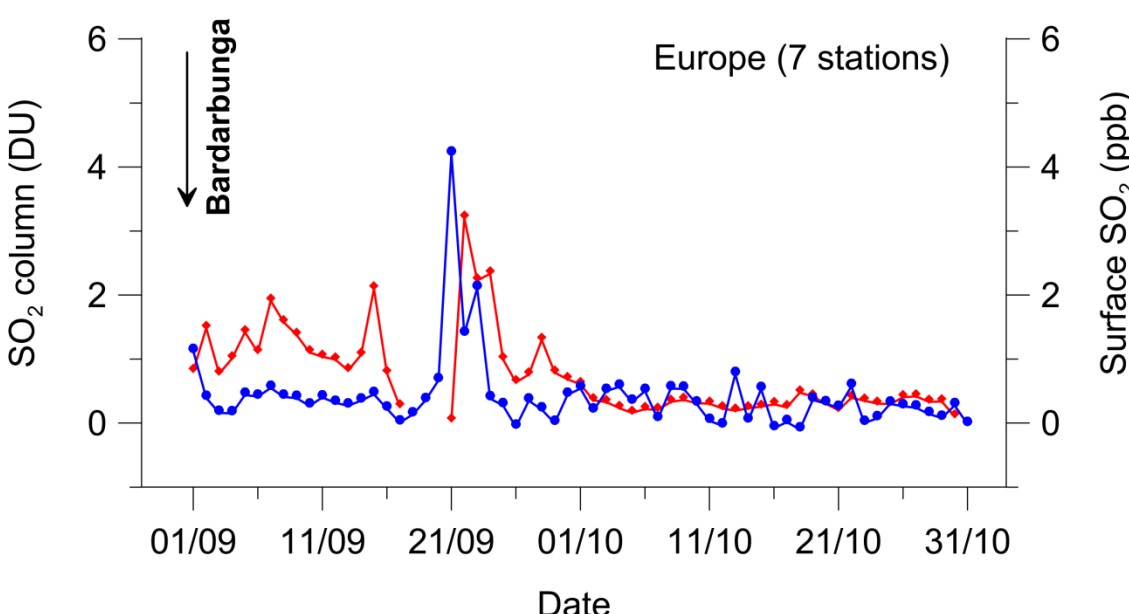

**Figure 5. Mean surface SO₂ measured by Airbase class 1-2 stations located within 150 km from 7 nearest Brewer stations in Europe as listed in Table 3. The arrow marks the starting date of the eruption (beginning on 31 August 2014 which continued to be active throughout the whole bimonthly period and beyond).**

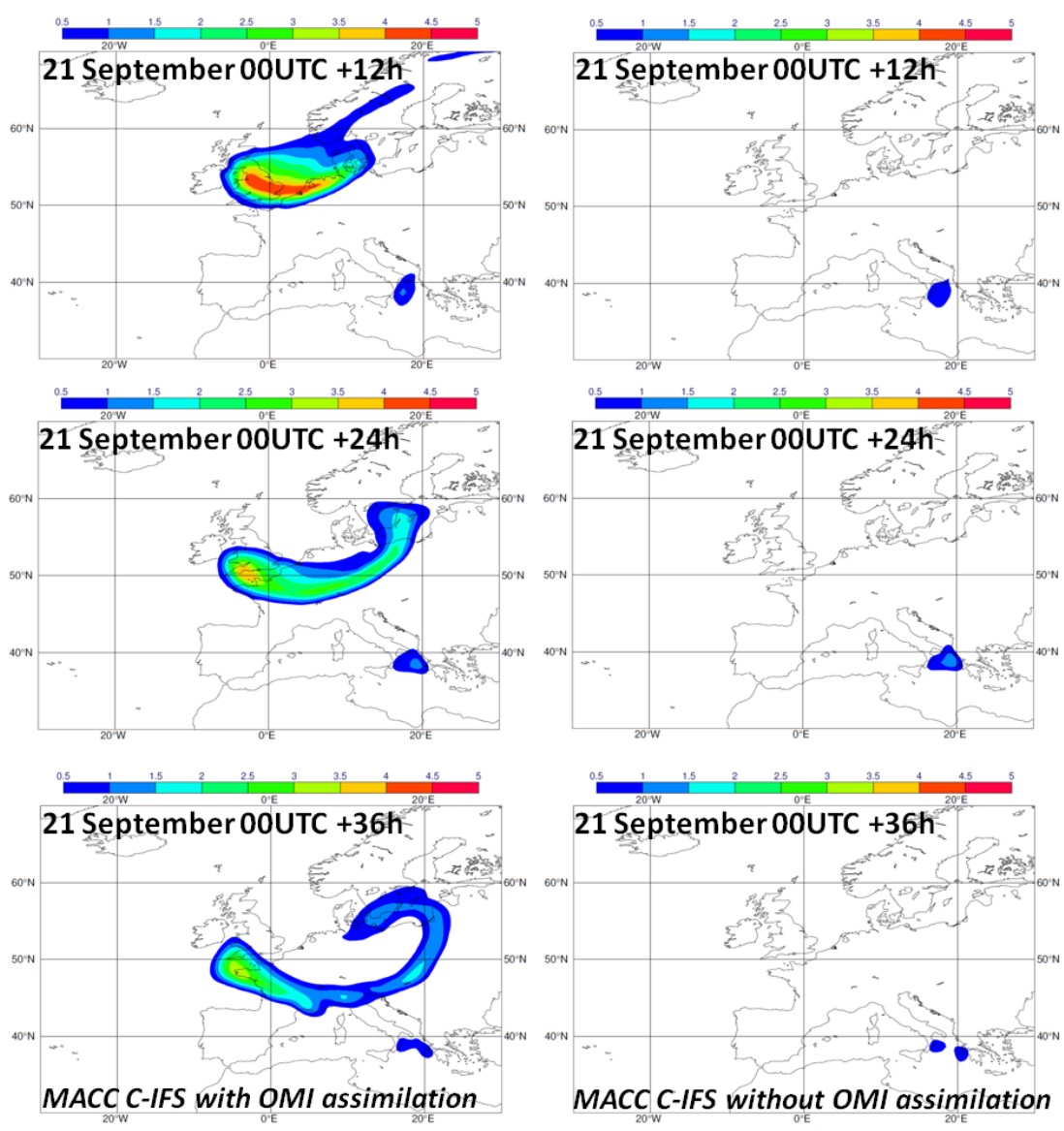

**Figure 6. Charts of forecasted total column SO$_2$ produced within the MACC system for 21 September 2014 with OMI data assimilation (left) and without OMI data assimilation (right).**

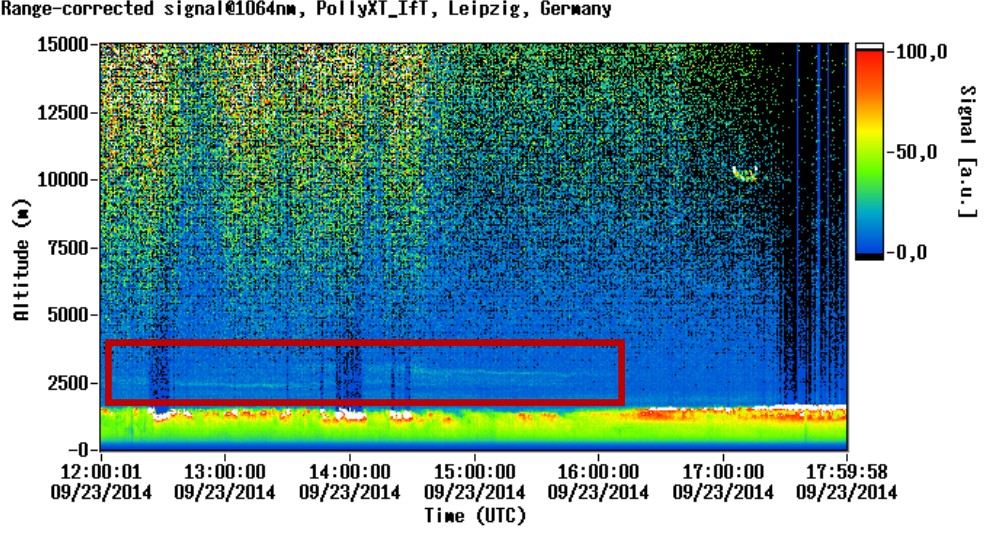

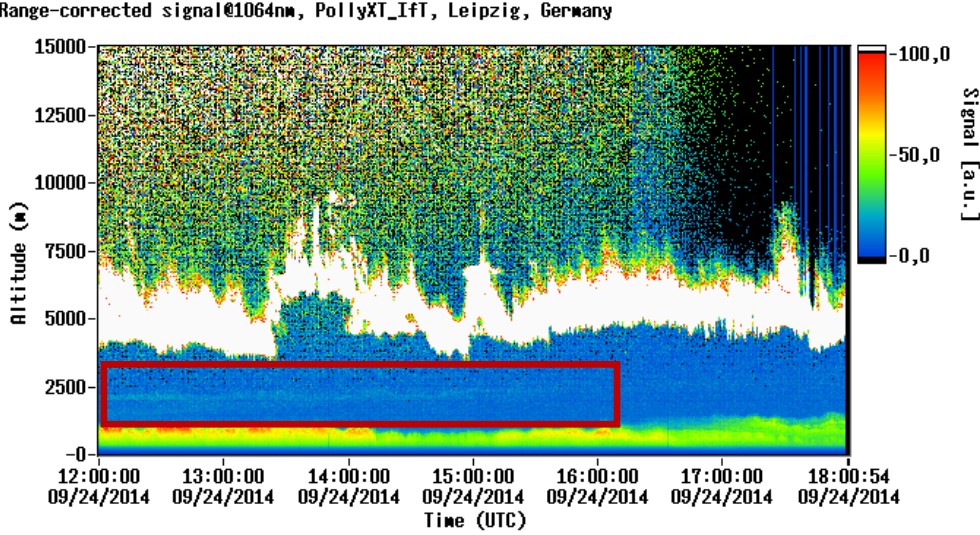

5  **Figure 7. Range corrected signal at 1064 nm from the PollyXT lidar in Leipzig on 23 September 2014 (up) and 24 September 2014 (down). The red rectangular indicates the location of the volcanic ash layer.**

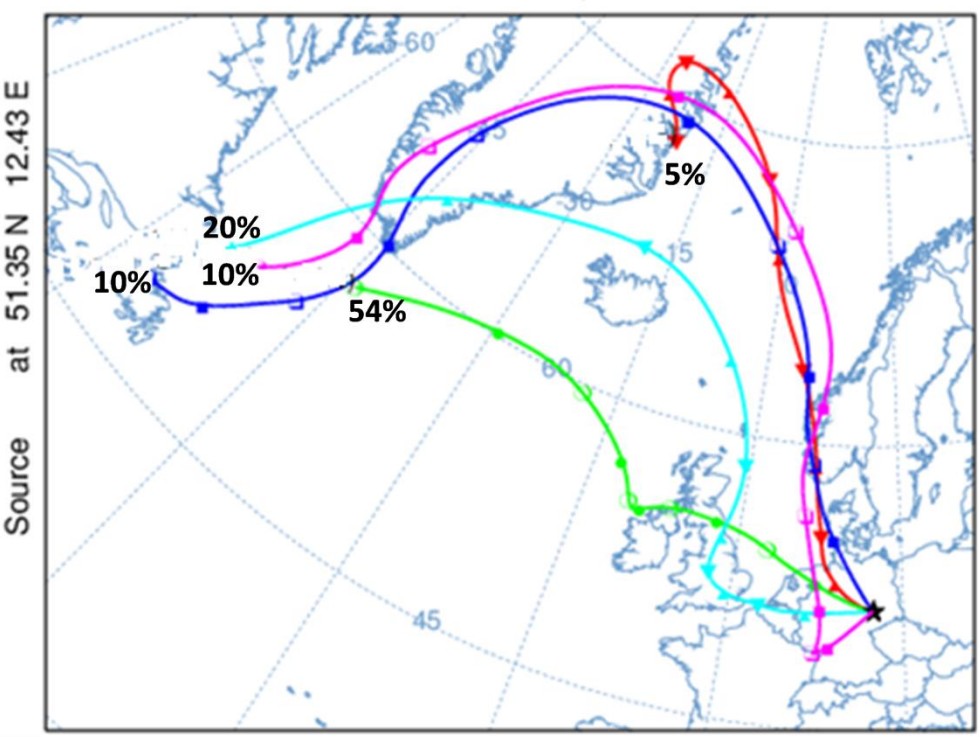

**Figure 8. Cluster analysis of the HYSPLIT back trajectories that arrive every hour (from 23 September 12:00 UTC up to 24 September 18:00 UTC) at 2.5-3.5 km height over Leipzig. A 54% cluster percentage means that there is 54% chance that the $SO_2$ arriving anywhere between 2.5-3.5 km over Leipzig originates from the specific direction.**

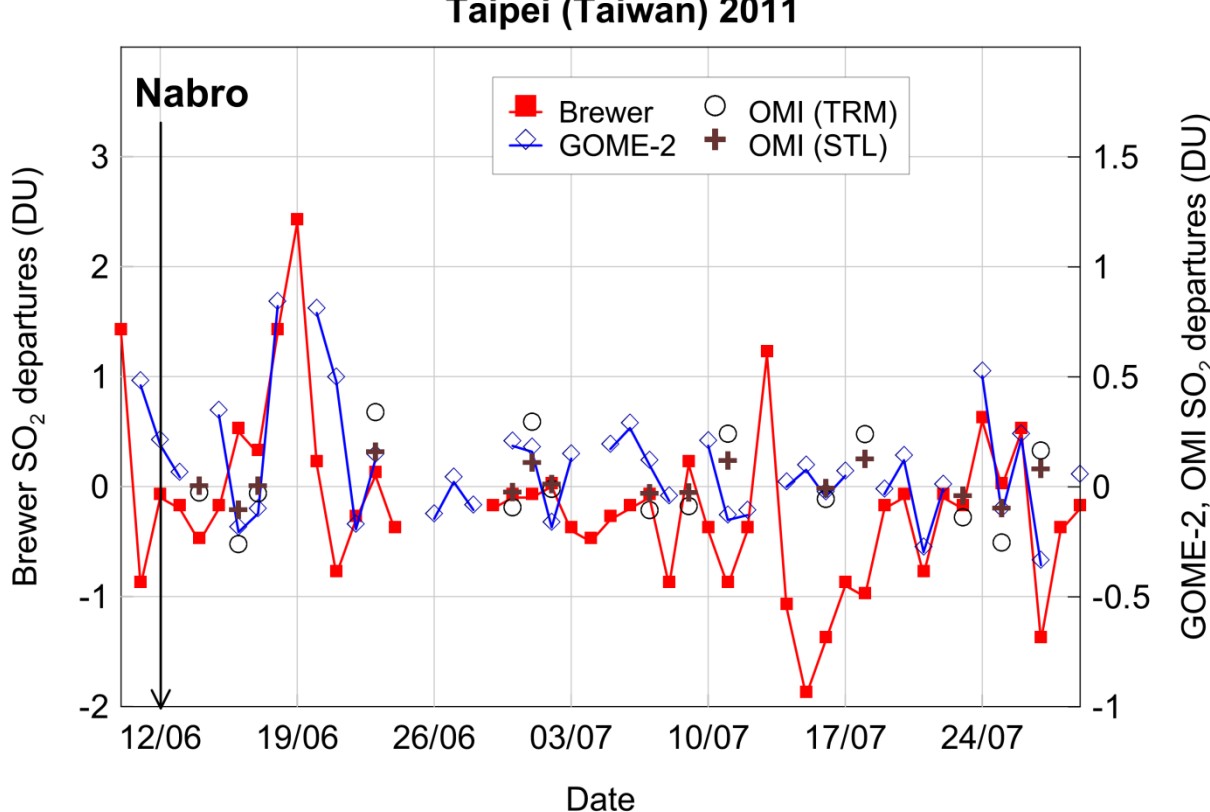

**Figure 9. SO₂ column departures from the unperturbed 10 day pre-volcanic baseline measured by Brewer, OMI (TRM, STL) and GOME-2 over Taipei, Taiwan, during June-July 2011 following the 2011 Nabro volcanic eruption (on 12 June 2011).**

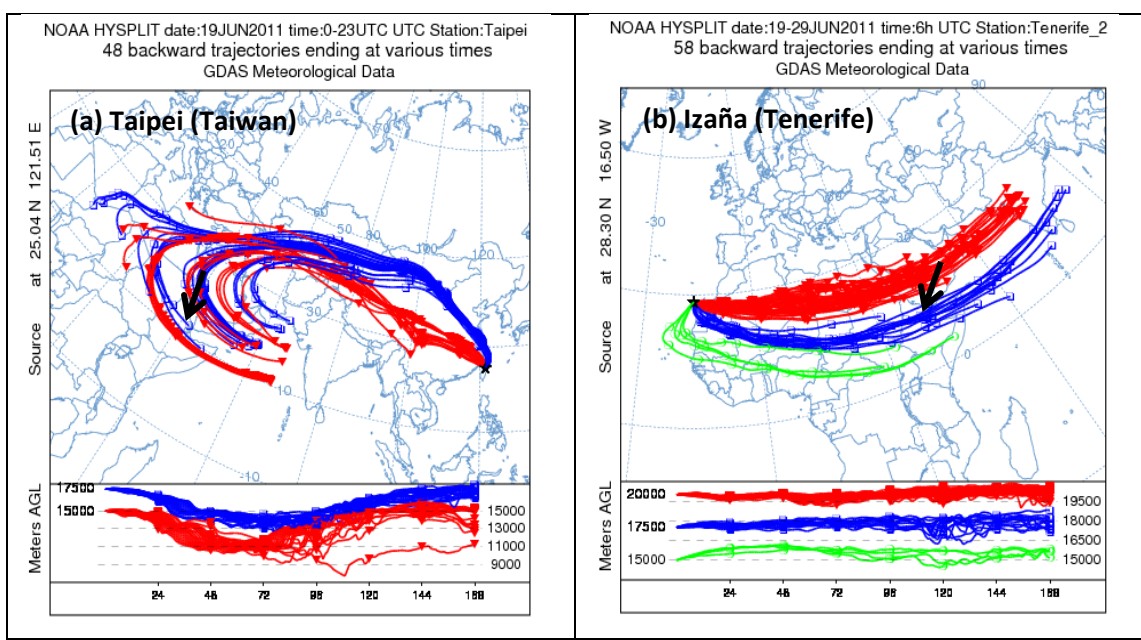

**Figure 10. HYSPLIT back trajectories of air masses (a) from Taipei (Taiwan) on 19 June 2011, (b) from Izaña (Tenerife) for days 19-29 June 2011. Nabro's location is indicated by the black arrow.**

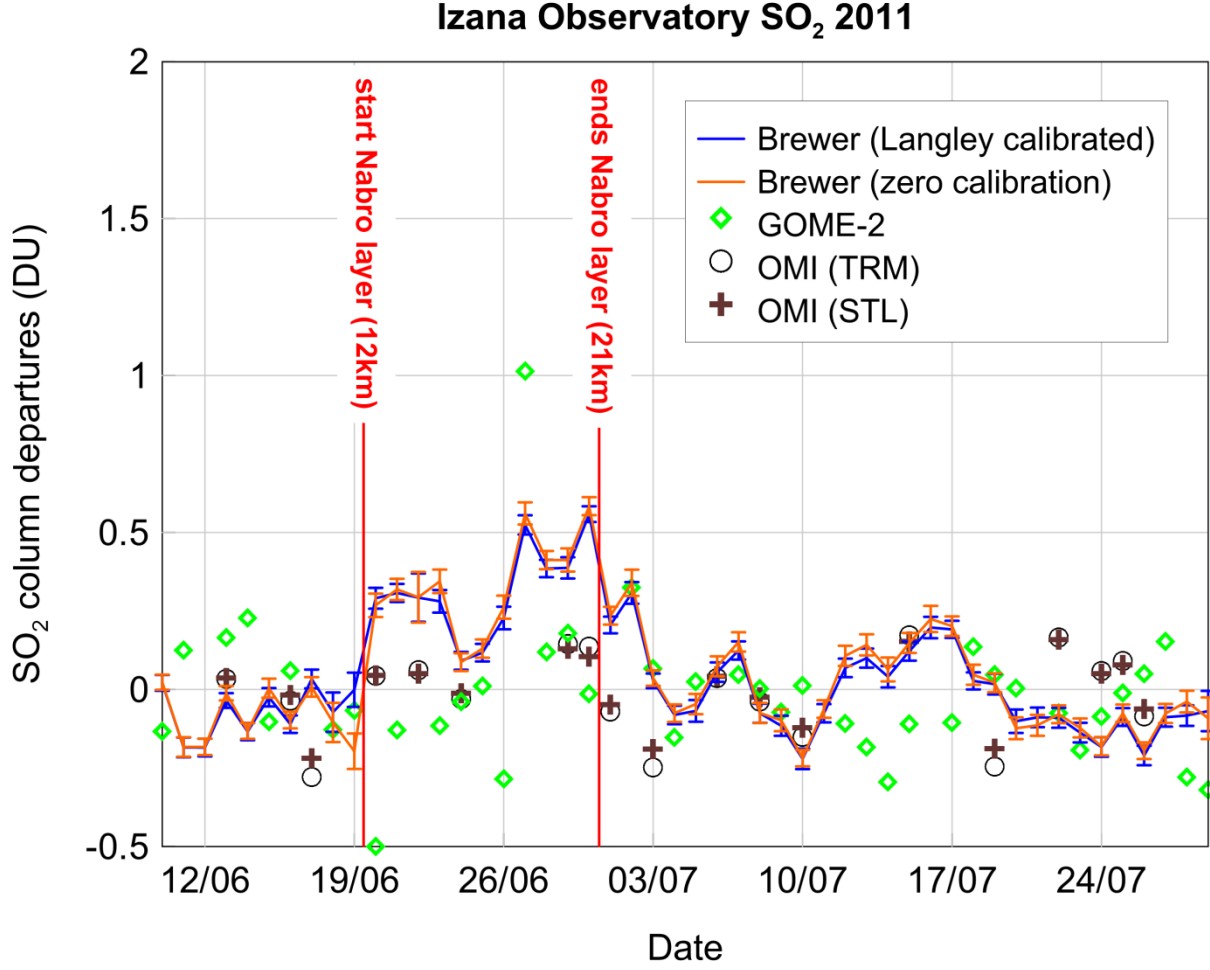

**Figure 11. SO₂ column departures from the unperturbed 10 day pre-volcanic baseline measured by the Brewer, OMI (TRM, STL) and GOME-2 over Izaña, Tenerife, during June-July 2011 following the 2011 Nabro volcanic eruption. SO₂ calculations by the Brewer were performed using the Langley calibration and the zero calibration at Izaña (assuming SO₂=0 during the days 06 and 07 of June 2011).**

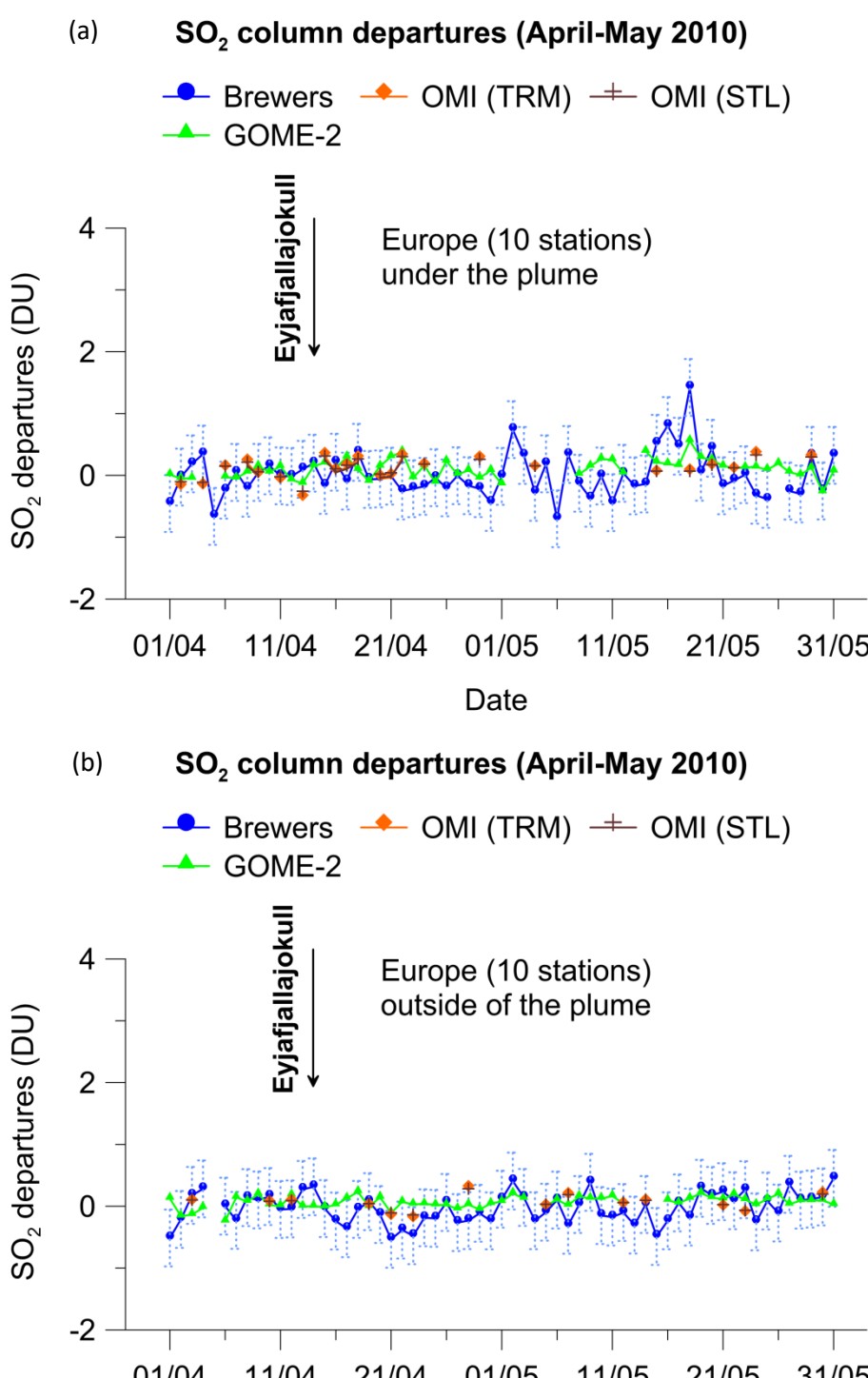

**Figure 12. Mean SO₂ column departures from the unperturbed 10 day pre-volcanic baseline measured by Brewers, OMI (TRM, STL) and GOME-2 during April-May 2010 over Europe following the 2010 Eyjafjallajökull volcanic eruption for: (a) stations under the volcanic SO₂ plume, and (b) stations outside of the plume. The error bars for the Brewer observations show the standard deviation of all daily values during the unperturbed 10 day period prior to the volcanic eruption. Brewer stations under the plume are: Sodankylä, Obninsk, Manchester, De Bilt, Uccle, Belsk, Reading, Hohenpeissenberg, Davos and Arosa. Stations outside of the plume are: Vindeln, Oslo, Norrkoeping, Copenhagen, Hradec Kralove, Aosta, Kislovodsk, Rome, Thessaloniki and Athens. Each daily average from either OMI or GOME-2 was calculated if and only if more than half of the individual overpasses had data at a given day. The arrow marks the starting date of the eruption (beginning on 14 April until 24 May 2010).**

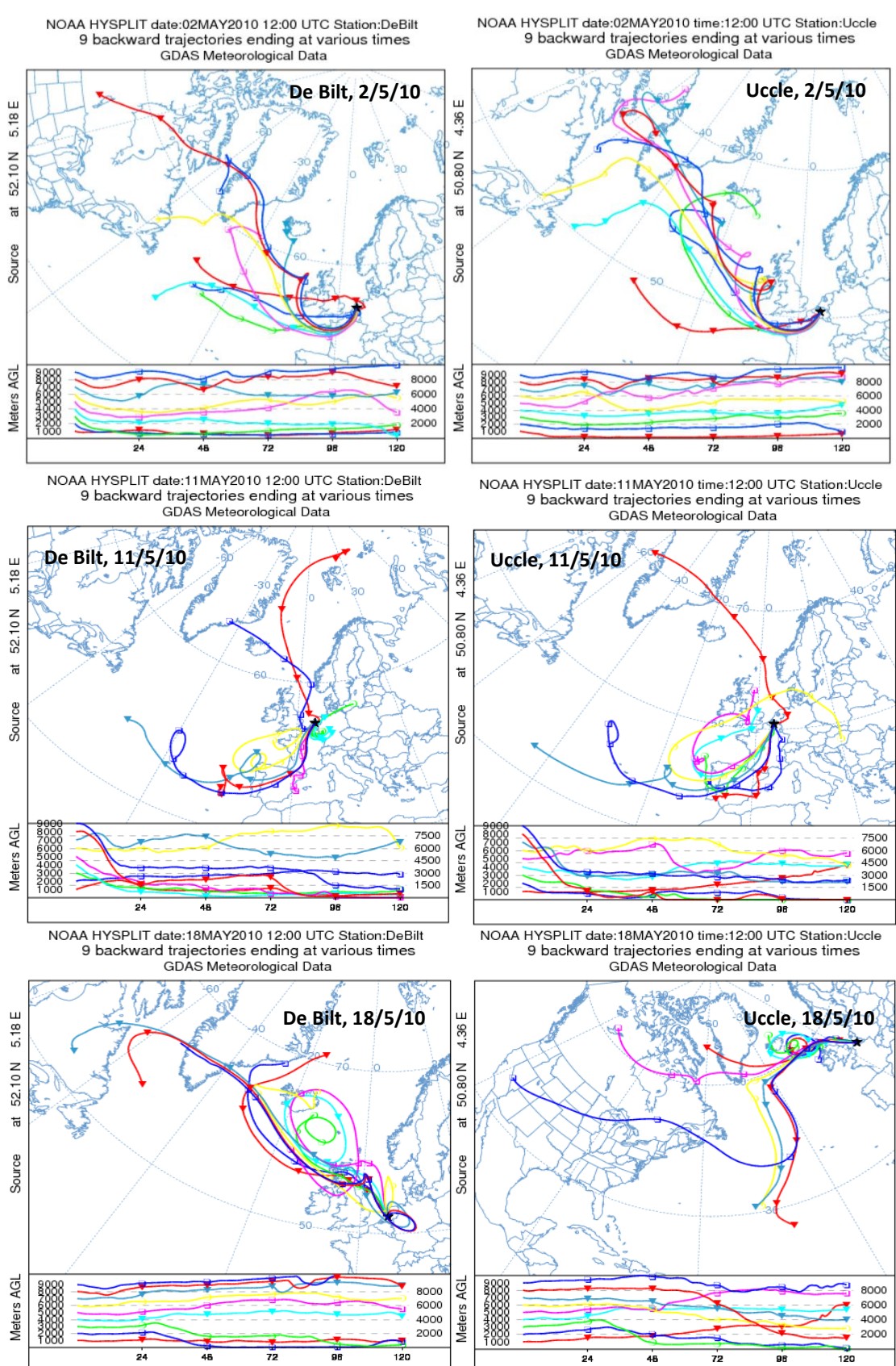

**Figure 13. HYSPLIT 120 hours back trajectories of air masses arriving at De Bilt (left column) and Uccle (right column) on 2 May 2010 (1st row), 11 May 2010 (2nd row), 18 May 2010 (3rd row).**

## SO$_2$ column departures (August-September 2008)

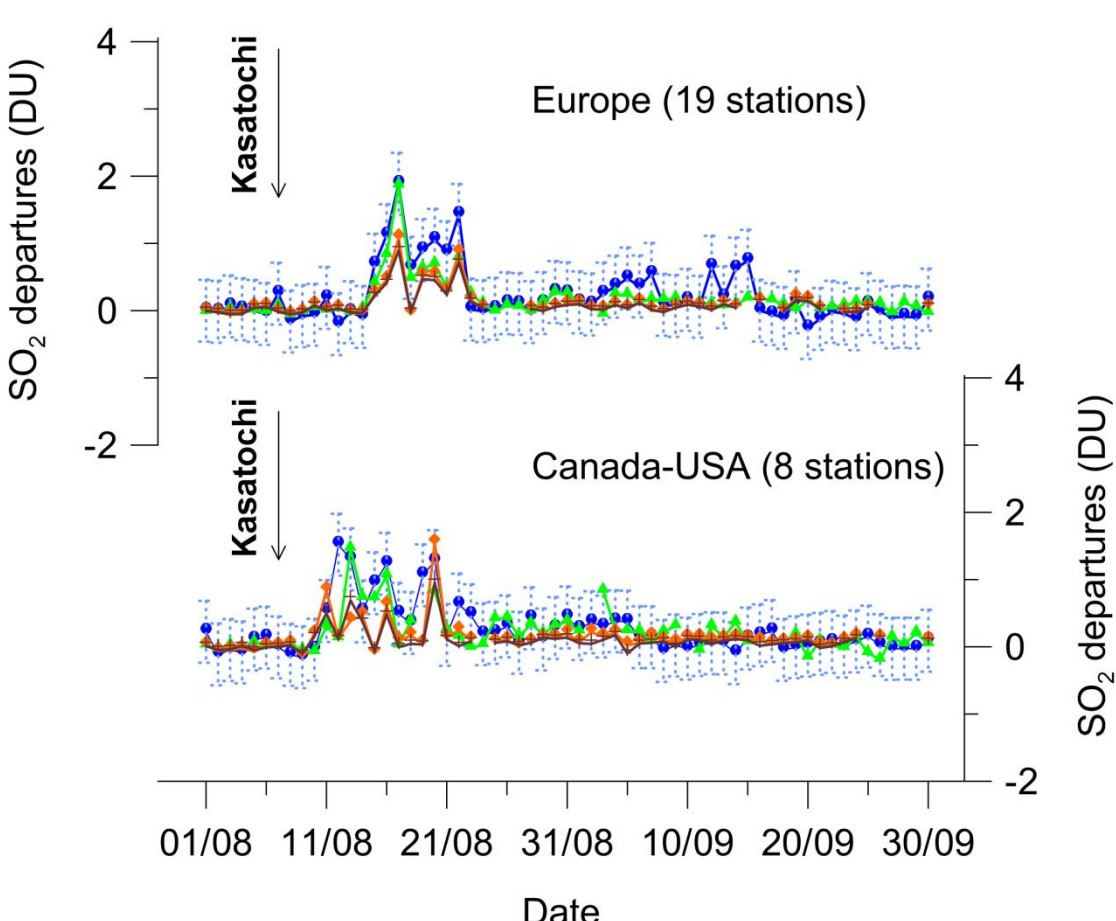

Figure 14. Mean SO$_2$ column departures from the unperturbed 10 day pre-volcanic baseline measured by Brewers, OMI (TRM, STL) and GOME-2 during August-September 2008 over Europe and Canada/USA following the 2008 Kasatochi volcanic eruption. The error bars for the Brewer observations show the standard deviation of all daily values during the unperturbed 10 day period prior to the volcanic eruption. Stations in Europe include: Sodankylä, Vindeln, Jokioinen, Norrkoeping, Copenhagen, Manchester, De Bilt, Belsk, Reading, Uccle, Hradec Kralove, Hohenpeissenberg, Davos, Arosa, Aosta, Kislovodsk, Rome, Thessaloniki and Athens. Stations in Canada/USA include: Churchill, Edmonton, Goose Bay, Regina, Saturna Island, Toronto, Boulder and Mauna Loa. Each daily average from either OMI or GOME-2 was calculated if and only if more than half of the individual overpasses had data at a given day. The arrow marks the date of the eruption (7 August 2008).