# Peer review of "Detecting volcanic sulfur dioxide plumes in the Northern Hemisphere using the Brewer spectrophotometer, other networks, and satellite observations"

_Atmospheric Chemistry and Physics, 2016_

## Referee Comment (RC1) · Anonymous Referee #1 · 28 Jul 2016

This paper investigates Brewer measurements following 5 volcanic eruptions since 2005 searching for an so2 signal in columnar spectrophotometric measurements. In all volcanic eruptions shown, a signal is found at the right time for the stations determined to be under the plume, based on trajectory modeling. In comparison no such signal is observed in those stations which are not under the plume based on the trajectories. By itself this is reasonably covincing of the reliability of the Brewers to detect so2 plumes from somewhat significant volcanic eruptions. This results is, however, tempered by the noisey so2 signal from the Brewers, and by difficulties in corroborating the Brewer so2 signal using satellite data, particularly the OMI data. The authors consistently

overlook this difficulty, making only a modest attempt to explain it. But then in the final case considered there is clear agreement between the Brewers, GOME-2, and OMI. The authors seize on this agreement, which is already clear, and preform a Pearson's statistical test leading them to claim in the conclusions, "The comparison with satellite measurements shows statistically tested agreement between the Brewer network and collocated measurements of columnar SO2 from OMI and GOME-2." But in fact this only applies to one of the eruptions considered, and if the same approach were used for the other four comparisons the conclusion would be significantly different. The authors need to temper their conclusions significantly and be more careful in explaining the comparisons.

While overall the results will be useful and the paper should be published there are some serious deficiencies which must be addressed. The overall tone in various places is disturbingly confident of the so2 measurements when there are glaring issues. The authors make claims starting with the first sentence of the abstract and ending in the conclusions (with the statement above), which are not correct. In addition there are several other serious issues which should be addressed.

The most serious issue is why there is such poor correlation of the satellite data, particularly OMI, with the Brewer data for 4 out of the 5 eruptions compared, and then such good agreement with Kasatochi? Was there something different about Kasatochi? When there is such poor agreement I don't see the point of quoting averages of the satellite data which appear to this reader to be in the noise of the measurments.

As the authors are aware, and I have become aware, the problem with so2 columns from Brewers is where to set the zero point. There appears to be a lot of noise around this value leading at times to negative so2 columns even in the satellite data. What do these negative columns mean? Can the authors provide a rough explanation of how to interpret them? Are they real or noise about a baseline of zero? Is a baseline determined for each of the instruments used?

[Figure]

On a similar theme, the non perturbed base line from, for example, the European Brewers has various values. The authors quote 0.46 DU as the mean of all the examined bi-monthly Brewer stations in contrast to means of -0.02 for OMI and 0.09 for GOME-2. How should this 0.46 DU be interpreted? Is this real or a bias in the data? If a bias should it be, or is it, subtracted? Do the authors know how the baseline is handled from their individual data sources and does it matter?

Considering Fig. 5, the 0.46 offset to the Brewer data looks reasonable as that appears to be about the non-perturbed baseline in all stations. But then in Fig. 10 for Taipei, the baseline appears in the 2-3 DU range and again OMI shows no correlation with the Brewer. The 0.46 DU offset also applies to Figs 14 and 15 b). In contrast in Fig. 15 a), c) the average is much closer to zero. What changed for these stations outside the plume, whereas within the plume the offset appears? Finally in Fig. 16 the offset appears to be near zero for the US/Canada and Taipei stations.

Aside from these major questions I provide the following specific comments, some provide more specifics on the themes above.

1.41-42. Have increased compared to what? That so2 columns increase following somewhat large volcanic eruptions is not new and has not depended on this paper to show that. Nor is it new that such columns increased following the five eruptions considered here. This sentence needs to be rephrased or deleted. I would begin the abstract with something like.

Following the five largest volcanic eruptions of the past decade in the Northern Hemisphere, a strong positive SO2 signal was detected by all the existing networks either ground based (Brewer, EARLINET, AirBase) or from satellites (OMI, GOME-2). This study particularly examines . . .

But after reading the paper even this sentence has issues. A strong signal was not detected in OMI and GOME-2 data according to the results shown here in several cases. Thus the statement that a ". . . a strong positive SO2 signal was detected by

all the existing networks either ground based (Brewer, EARLINET, AirBase) or from satellites (OMI, GOME-2) . . ." is not correct for the satellite data for all cases.

1.41. Why are the increases described as significant? Significant in what way? The so2 increases following Pinatubo and El Chichon were significant, but these are on a different scale than the eruptions considered here.

1.45-47. This statement is incorrect for the reasons given above, particularly for OMI. The correlation is better for Brewer and GOME-2, but I doubt even this would be statistically significant at the level claimed if all cases were considered. See Figs. 5, 12, 15. Again how are the columnar so2 amounts significant? What do the authors intend to imply with this word?

3.9-14. The authors need to be more careful about their claims concerning the "five" volcanic eruptions. In the abstract it was the 5 most significant eruptions since 2005. Now here it seems to be the five eruptions which produce the most so2 over Iceland, but only 4 eruptions are shown. Not surprisingly 3 of these eruptions were in Iceland, although most of these eruptions are not on the list of the 5 eruptions since 2005 with the greatest atmospheric impact. Here the sentence needs to indicate up front that these are selected based on their so2 columns over Iceland. So . . . Five cases of high SO2 over Iceland from volcanic . . .

Yet this sentence goes on to say that these are the five eruptions to be compared in this study. So I am confused, are the eruptions the 5 most significant since 2005 or the 5 with most significant so2 over Iceland. According to the Smithsonian Global Volcanism Network, BárÃřarbunga has a VEI of zero, so undetermined.

Table 1. It is significant that 4 of the 5 eruptions are at high northern latitudes, while the lone tropical eruption had its plume picked up in the Asian monsoonal circulation to bring the so2 over Europe, so an important but poorly stated criteria seems to be the opportunity to measure the plume over Europe.

[Figure]

Clearly there is enough confusion here that the authors need to rethink the criteria used for the selection of the 5 eruptions and to explain it clearly.

4.13-21. Confusing. I had to re-read this several times. First the authors state . . . the Brewer spectrophotometer is additionally used to derive the SO2 column.., Then the say . . . The existing Brewer network could deliver frequent SO2 measurements as well, but the Brewer instruments are less able to accurately provide SO2 measurements . . . So which is it? Don't claim that it is used and then say it can't be used. Please rewrite this to be clear.

7.2-4. Doesn't this also suggest a bias for the Brewer data?

7.35-36. From Fig. 5 only the GOME-2 measurements corroborate the Brewer results, but even then only in timing, not in magnitude. Is there an explanation why no signal appears in OMI data and why the Brewer and GOME disagree in magnitude to the extent shown?

8.23-30. Aside from GOME-2 it seems pointless to quote these numbers for OMI. The OMI data do not indicate anything out of the ordinary for 20-25 September, neither the TRM or PBL. In fact there are bigger excursions of the so2 column at other times. The GOME-2 data are better and a case can be made that some so2 was observed, but even these data could be questioned.

8.33-35. Thus the statement, "In all cases, however, the observed . . . were always higher . . ." is simply incorrect, as demonstrated with the numbers just above, and should be removed.

9.1-5. Why is there so much inconsistency between Figures 5 and 7. Fig. 7 shows OMI measurements of 1-4 DU across large regions of Europe, yet Fig. 5 indicates almost all OMI measurements < 1 DU and most measurements < 0.5 DU.

Figure 9. The differences between the colored lines are not obvious.

10.15. What is meant by both methods?

Fig. 15. Why is the Brewer baseline at 0.2-0.3 DU for the stations under the plume, whereas for the 10 outside stations the baseline is closer to zero?

11.38. Does an average $SO_2$ plume of 0.1 DU mean anything when earlier the averages of the Brewers without influence by volcanoes was on the order of 0.4 DU? It does not help the authors' argument to be calling out numbers in the text which are in the noise of the measurements. The authors also never explain what a negative DU measurement means. What causes this? Are the negative numbers a real measurement?

Fig. 16. Why is a 7 day running mean now added to the measurements? Does it show something missing in the simple averaged daily data shown up to now?

12.31-13.6. A calculation of Pearson's correlation coefficients is not necessary to convince the readers that the Brewers, GOME-2 and OMI are all in agreement at least over Europe. Is the Taiwan station included in the correlation coefficients? If so, does the fact that there is virtually no correlation there get masked because it is only one station? What is telling about this paragraph, and the corresponding Table 5, is that such tests were not used in any previous comparison, most certainly because the results would have been much worse, see Figures 5, 12, 15.

13.16-18. This statement is based on only the Kasatochi results and does not hold for 4 of the 5 eruptions studied, thus the statement either has to be removed from the conclusions or dampened considerably by pointing out all the other times when no correlation or a poor correlation was found.

---

## Referee Comment (RC2) · Anonymous Referee #2 · 1 Aug 2016

This article provides an interesting assessment of the detection of volcanic sulfur dioxide by Brewer spectrometers in the Northern hemisphere. It uses surface station and satellite measurements as well as trajectory models to evaluate the performance of the Brewer instruments in case of elevated SO2 levels due to the passage of volcanic plumes. Although the Brewer instruments are not accurate enough to monitor SO2 on the long term, it is argued, by using trajectory models in order to trace large (VEI>4) volcanic plumes from recent eruptions in the Northern hemisphere, that Brewer instruments can detect the volcanic SO2 signals. Since the Brewer network set up for the monitoring of total ozone includes a large number of stations, the authors suggest

to use this measurement capability to forecast the evolution of volcanic plumes and provide a new SO2 forecasting tool. The paper is correctly written and informative regarding SO2 measurement capacity of Brewer instruments. I recommend publication in ACP, provided that important comments for improvement are taken into account.

Main comments

• The measurement capability of Brewer instruments should be better explained. Since the paper focuses on the detection of small SO2 signals, the methodology to derive SO2 total content should be summarized in the paper itself. An assessment of the mean SO2 values generally provided by Brewer instruments should be provided.

• As optical instruments, the Brewer measurements can be perturbed by ash present in the volcanic plumes. This issue should be addressed in the article.

• For readers not familiar with total SO2 measurements by Brewer spectrometers, it is rather intriguing to see negative total SO2 values. So it would be worth explaining why such negative values have to be considered in the general Brewer (and satellite) retrieval.

• Two lagrangian models are used for the analysis: FLEXPART and HYSPLIT. An explanation is needed on why two different models need to be used (paragraph 2.3).

• In the case of the Baraorbunga volcano, the FLEXPART model has been used to simulate SO2 levels in air masses sampled at Hohenpeissenberg station. But there is no detail on the simulation and on the initial emitted SO2 levels.

• For the same volcano, it is not completely clear that the elevated SO2 levels detected by ground stations correspond to the volcanic plume. Also a better explanation should be given on why the plume is not seen in OMI and GOME 2 measurements shown in Figure 5. The case for the detection of this volcanic plume by the satellite instruments over Europe and for the attribution of increased SO2 levels from these measurements (page 8) is not completely made.

• The fact that the 2011 Grimsvöth volcanic plume was not detected by the European Brewer instrument does not bring much to the article. This paragraph should be removed.

• Again for the Eyjafjallajökul volcano, OMI and GOME 2 do not seem to detect the SO2 signal. An explanation is needed on the lack of detection by satellite instruments. Also, the left panel of Figure 16 is redundant with the right panel.

• 2008 Kasatochi case: it is not clear from the article why the plume is not detected in Taïwan by the satellite instruments, contrary to the observations in Europe and North America. This issue should be addressed.

• The conclusion should better summarize in which general conditions (SO2 levels, time after eruption) Brewer instruments can be useful for the detection of SO2 volcanic plumes. The article is qualitative in general and such a summary would provide a quantified assessment of the measurements capability of Brewer instruments with respect to SO2 measurements. Comparison with OMI and GOME 2 measurements capacity in similar cases would be useful. It would be also worth mentioning why IASI and AIRS measurements are not included in the analysis.

Minor comments

In general, figures' legends should be more informative, with the description of the various plots and the name of the volcano case to which the figure refer (when SO2 levels are plotted).

Figure 7: can the authors comment on the spot of elevated SO2 observed between Italy and Greece?
* * *

---

## Author Response (AR1)

**Reply to Reviewer #1**

We are thankful to Reviewer #1 for the constructive comments. His introductory remarks/questions have been grouped as 3 general questions. To these 3 general questions, as well as to his additional 17 specific comments, our responses and revisions in the text are as follows:

**General question 1: Reviewer #1 criticizes our overlooking of the difficulties to see particularly in the OMI data the volcanic $SO_2$ signals seen by the Brewers and as he points out "The authors need to temper their conclusions ....".**

**Response to general question 1:** Reviewer #1 correctly points out that we should have tempered our conclusions concerning the SO2 excursions following large volcanic eruptions because they could not be seen equally well in the OMI and GOME-2 satellite measurements as was the case with the Brewer network, except for Kasatochi. We have carefully revisited the OMI and GOME-2 data sets and found out that during the most perturbed period following the eruptions of Bardarbunga and Eyjafallajökull the satellite measurements from overpasses were so sparse that the daily average was not corresponding to the Brewer network sample. For instance and following Bardarbunga and Eyjafjallajökull, there were many days where we had only one or two OMI overpassing measurements following the eruption, obviously not representing the 19 Brewer instruments in Europe. To temper our past conclusions we have applied a criterion (see new section 3.1) according to which "a daily average from either OMI or GOME-2 should be calculated if and only if more than half of the individual overpasses had data at a given day". As can be seen from the revised figures 4 and 12, OMI data are missing for not meeting this criterion. The only firm conclusion that can be drawn with statistical confidence is that from all three eruptions with volcanic SO2 plumes overpassing the Brewer network and seen as well from OMI and GOME-2, a strong positive signal can be confirmed only in the case of Kasatochi eruption (we have redrawn the time series, see new Fig. 13). Following these major changes, we have rephrased our abstract and conclusions accordingly.

**General question 2: "The most serious issue is why there is such poor correlation of the satellite data, particularly OMI, with the Brewer data for 4 out of the 5 eruptions compared, and then such good agreement with Kasatochi? Was there something different about Kasatochi? When there is such poor agreement I don't see the point of quoting averages of the satellite data which appear to this reader to be in the noise of the measurements".**

**Answer to general question 2:** Indeed as mentioned above the best agreement was found for the case of Kasatochi because it happened to have many measurements from coinciding satellite overpasses during common days with the Brewer instruments. For the case of Bardarbunga and for the case of Eyjafjallajökull, the satellite data were sparse, particularly for OMI. For Bardarbunga, the correlation between the GOME-2 overpasses and Brewer stations under the volcanic SO2 plume was calculated to be 0.44, statistically significant at the 99% confidence level in spite the fact that during the two days of peak SO2 levels (21-22/9/2014) as "seen" at the Brewer stations, there were no satellite data available. For Eyjafjallajökull similar sparsity of the data reduces confidence and unfortunately for OMI we could not calculate correlations with the Brewers at all due to the small sample of the satellite data. We note here that the case for Grimsvotn volcano has been removed as recommended by reviewer #2 comments and is not discussed in the revised paper. The

reason is that the volcanic  $SO_2$  plume has been always outside of the Brewer network. The text has been revised in concurrence to the above findings.

**General question 3: The reviewer points out the problem in measuring $SO_2$ columns, where to set the zero point as well as what is the meaning of negative $SO_2$ columns and how to interpret them and related questions on the noise, the baseline and the correlations in figures 5, 10, 14, 15 and 16.**

**Answer to general question 3:** In the text (section 2.1) we have added a full description of the Brewer algorithm and the reasoning on the existence of some negative values which could be considered either as small or as noise. The text now reads: "From the above described operational Brewer algorithm it is evident that the estimation of columnar SO2 is the result of the difference between two columnar terms  $(O_3 + SO_2)$  and  $O_3$ . Both terms have uncertainties (weighting functions, calibrations, random errors, systematic errors). Systematic negative values could be the result of a systematic offset in the measurements that can be related to the calibration of the instrument (usually optimized only for the ozone measurements). Randomly varying positive and negative values around zero, suggest that the signal of SO2 is small (and thus the difference of two terms should be close to zero) but since both terms have uncertainties, negative values are possible indicating that the amount of SO2 in the atmosphere is below the detection limit of the instrument and could be considered as noise. In this work we have repeated our analysis excluding the negative values and the results remained the same i.e. a positive increase after a major volcanic eruption was confirmed as described in the following sections".

After careful consideration, we decided to recalculate all values and redraw all Brewer composite figures by considering that 10 days before the volcanic eruption all Brewer and satellite observations obviously did not contain any volcanic signal. The data set which included daily values during the 10-day unperturbed period before the eruption, was considered to represent the base line for each Figure. Subsequent grouping in the **new Figures** (4, 9, 12, 13) show the departures of mean SO2 columns from the unperturbed baseline and all numbers in Table 4 have been recalculated as departures from the unperturbed 10-day baseline.

**Answers to specific comments**

Comment 1: "1.41-42. Have increased compared to what? That so2 columns increase following somewhat large volcanic eruptions is not new and has not depended on this paper to show that. Nor is it new that such columns increased following the five eruptions considered here. This sentence needs to be rephrased or deleted. I would begin the abstract with something like.

Following the five largest volcanic eruptions of the past decade in the Northern Hemisphere, a strong positive  $SO_2$  signal was detected by all the existing networks either ground based (Brewer, EARLINET, AirBase) or from satellites (OMI, GOME-2). This study particularly examines ...

But after reading the paper even this sentence has issues. A strong signal was not detected in OMI and GOME-2 data according to the results shown here in several cases. Thus the statement that a "... a strong positive  $SO_2$  signal was detected by all the existing networks either ground based (Brewer, EARLINET, AirBase) or from satellites (OMI, GOME-2) ..." is not correct for the satellite data for all cases". **Answer to comment 1**: In the revised text we clarify that the  $SO_2$  columns have increased relative to the unperturbed 10-day baseline. We also specify that a strong positive signal was detected by all the existing networks only at Kasatochi. As mentioned before, the abstract and conclusions have been fully revised accordingly.

**Comment 2: "1.41. Why are the increases described as significant? Significant in what way? The so2 increases following Pinatubo and El Chichon were significant, but these are on a different scale than the eruptions considered here".**

**Answer to comment 2:** In the revised text the increases are described as departures from the ten days before the eruption where all Brewer and satellite  $SO_2$  measurements are considered as non-perturbed. A departure was characterised significant if it exceeded  $3\sigma$ , where  $\sigma$  was calculated from all daily values 10 days before all eruptions and for as many locations as the number of the measuring stations or the corresponding satellite overpasses in the cases of OMI and GOME-2.

Comment 3: "1.45-47. This statement is incorrect for the reasons given above, particularly for OMI. The correlation is better for Brewer and GOME-2, but I doubt even this would be statistically significant at the level claimed if all cases were considered. See Figs. 5, 12, 15. Again how are the columnar so2 amounts significant? What do the authors intend to imply with this word?"

**Answer to comment 3:** In our original manuscript sparsity of data from OMI and to a lesser extent from GOME-2 resulted to wrong correlations with the data from the Brewers. In the revised text the correlations between the Brewers and GOME-2 have been corrected and were estimated to be 0.31 (95% confidence level) and 0.44 (99% confidence level) in Eyjafjallajökull and Bárðarbunga, respectively. Correlations between the Brewers and OMI were not calculated due to the scarcity of OMI data in Eyjafjallajökull and Bárðarbunga (see corrected Table 5, corrected text and abstract).

Comment 4: "3.9-14. The authors need to be more careful about their claims concerning the "five" volcanic eruptions. In the abstract it was the 5 most significant eruptions since 2005. Now here it seems to be the five eruptions which produce the most so2 over Iceland, but only 4 eruptions are shown. Not surprisingly 3 of these eruptions were in Iceland, although most of these eruptions are not on the list of the 5 eruptions since 2005 with the greatest atmospheric impact. Here the sentence needs to indicate up front that these are selected based on their so2 columns over Iceland. So ... Five cases of high SO2 over Iceland from volcanic ...

Yet this sentence goes on to say that these are the five eruptions to be compared in this study. So I am confused, are the eruptions the 5 most significant since 2005 or the 5 with most significant so2 over Iceland. According to the Smithsonian Global Volcanism Network, Bárdarbunga has a VEI of zero, so undetermined.

Table 1. It is significant that 4 of the 5 eruptions are at high northern latitudes, while the lone tropical eruption had its plume picked up in the Asian monsoonal circulation to bring the so2 over Europe, so an important but poorly stated criteria seems to be the opportunity to measure the plume over Europe.

**Clearly there is enough confusion here that the authors need to rethink the criteria used for the selection of the 5 eruptions and to explain it clearly".**

**Answer to comment 4:** We consider all major eruptions that have occurred in the N.H. in the past decade according to the Smithsonian Global Volcanism. The text has been revised and reads now as follows:

"Table 1 lists in chronological order all major volcanic eruptions in the Northern Hemisphere between 2005-2015 with volcanic explosivity scale index (VEI) of at least 4 (Newhall and Self, 1982; Robock et al., 2000; Zerefos et al., 2014). The study also provides a separate analysis for the Bárðarbunga eruption, which although not rated 4 has been already studied with the Brewer at Sodankylä by Jalongo et al. (2015).

As seen from Table 1, chronologically, the first case was the volcanic eruption at Mount Okmok, Alaska (53.43° N, 168.13° W, 1073 m above sea level (asl), 12 July 2008, Prata et al., 2010) followed by the Kasatochi eruption, Alaska (52.17° N, 175.51° W, 300 m asl, 7-8 August 2008, e.g., Kristiansen et al., 2010; Krotkov et al., 2010; Waythomas et al., 2010) which was detected over large areas of the Northern Hemisphere. Okmok and Kasatochi volcanoes in Alaska erupted a short time span of less than a month and therefore we decided to study the evolution of the Brewer SO2 columnar measurements following the latter volcanic eruption (Kasatochi). The third eruption took place at Sarychev in Russia (48.1° N, 153.2° E, 1496 m asl, 12-17 June 2009, Haywood et al., 2010). The evolution of the SO2 volcanic plume from Sarychev was mostly observed over the North Pacific, North America and North Atlantic (Haywood et al., 2010). There was only one North American Brewer station (Saturna Island) in the path of the plume from Sarychev eruption. The record shows SO2 columns of 8.6 DU detected on 19 June 2009 and 3.7 DU on 20 June 2009. This volcanic eruption is not investigated any further in this paper. The next eruption on the list, Eyjafjallajökull in Iceland (63.63° N, 19.62° W, 1666 m asl, from 14 April to 23 May 2010), resulted in interruption of the air traffic over NW Europe (e.g. Flemming and Inness, 2013). The fifth eruption Grímsvötn 2011 (64.42° N, 17.33° W, 1725 m asl, 21 May 2011) was studied by Flemming and Inness (2013), and by Moxnes et al. (2014). This eruption provided an interesting example of a clear separation of the volcanic  $SO_2$  plume (transported mostly northwestward) while the fine ash was transported mostly southeastward. Unfortunately the volcanic plume did not overpass any Brewer station and therefore we do not include any results post Grímsvötn eruption. The sixth eruption recorded features the Nabro in Africa (13.37° N, 41.70° E, 2218 m asl) that occurred on 12-13 June 2011 (e.g., Bourassa et al., 2012; Sawamura et al., 2012; Clarisse et al., 2014). We present here a case study that described detection of the Nabro volcanic SO2 plume over ground based stations. The plume was clearly detected by the Brewer instrument over Izaña (and poorly from space), then over Taiwan by both Brewer and satellite instruments, and finally at Mauna Loa, Hawaii (mostly by the Brewer instrument). The seventh eruption was Tolbachik, Russia (55.83° N, 160.33° E, 3.611 m asl) on 27 November 2012 (e.g. Telling et al., 2015). As in the case of Grímsvötn, the plume has not passed over any Brewer station that was verified by trajectory analysis. The next eruption on the list is the volcanic eruption from Bárðarbunga, Iceland (64.64° N, 17.56° W, 2005 m asl) that was observed between 31 August 2014 and 28 February 2015 (e.g. Schmidt et al., 2015). This last eruption, although not yet rated on the VEI scale, has been extensively studied in view of the observed increased SO2 concentrations that have been observed all the way through troposphere and reaching down to the surface in Europe (Ialongo et al., 2015; Schmidt et al., 2015)."

Comment 5: "4.13-21. Confusing. I had to re-read this several times. First the authors state ... the Brewer spectrophotometer is additionally used to derive the SO2 column.., Then they

**say ... The existing Brewer network could deliver frequent $SO_2$ measurements as well, but the Brewer instruments are less able to accurately provide $SO_2$ measurements ... So which is it? Don't claim that it is used and then say it can't be used. Please rewrite this to be clear".**

**Answer to comment 5:** The sentence has been rewritten and reads as follows: "Because sulfur dioxide has strong and variable absorption in this spectral region, the Brewer spectrophotometer has additionally been proposed to derive  $SO_2$  columns (Kerr et al., 1980). About two hundred Brewer spectrophotometers around the world contribute high-precision ozone data to the global ozone monitoring network (Kumharn et al., 2012). The existing Brewer network also delivers frequent  $SO_2$  columnar measurements as well, which can be used for analyses, but with caution". (See revised section 2.1).

**Comment 6: "7.2-4. Doesn't this also suggest a bias for the Brewer data?"**

**Answer to comment 6:** Any biases in the data have been eliminated by expressing all data (Brewer, GOME-2 and OMI) as departures from the unperturbed 10 day period prior to the volcanic eruptions. The new text now reads: "Averaging the departures from the prevolcanic baseline for all Brewer stations and for all bimonthly periods gives a mean SO2 columnar departure of 0.10 ± 0.03 DU. This estimate is on the same order of magnitude as the corresponding statistics for OMI (TRM) SO2 column departures (0.05 ± 0.02 DU) and that measured by GOME-2 (0.09 ± 0.02 DU)".

**Comment 7: "7.35-36. From Fig. 5 only the GOME-2 measurements corroborate the Brewer results, but even then only in timing, not in magnitude. Is there an explanation why no signal appears in OMI data and why the Brewer and GOME disagree in magnitude to the extent shown?"**

**Answer to comment 7:** The explanation is the sparsity of OMI and GOME-2 data, particularly OMI, during the days of elevated SO2 column observed by the Brewer network. Figure 5 (new figure 4) has been redrawn by applying a criterion according to which a daily average from either OMI or GOME-2 should be calculated if and only if more than half of the individual overpasses had data at a given day. The text has been revised and reads now as follows: "As shown in Figure 4a, the SO2 plume was detected by the Brewer instruments located in the passage of the volcanic SO2 plume and from different ground based networks. However, no co-incident measurements were available from the OMI and GOME-2 overpasses at the time of the high SO2 excursions".

**Comment 8: "8.23-30. Aside from GOME-2 it seems pointless to quote these numbers for OMI. The OMI data do not indicate anything out of the ordinary for 20-25 September, neither the TRM nor PBL. In fact there are bigger excursions of the so2 column at other times. The GOME-2 data are better and a case can be made that some so2 was observed, but even these data could be questioned".**

**Answer to comment 8:** In the revised text we do not quote these numbers for OMI. The new text now reads: "As can be seen from Figure 4a, the highest SO2 column departures from the pre-volcanic baseline were observed from 21 to 22 September 2014. The mean SO2 column measured by the Brewers under the plume was  $2.4 \pm 0.8$  DU, which was five times greater

than the mean column of SO2 measured by the Brewers outside of the plume (-0.1  $\pm$  0.1 DU) by 2.5 DU on average. The "error bars" show the standard deviation of the daily SO2 values of all stations during the non-perturbed 10 day period prior to the volcanic eruption. These differences provide rough estimates of the additional SO2 loading induced by the volcanic eruption over Europe which exceeds  $3\sigma$ . Comparison between satellite data and Brewer are limited for interpretation because satellite measurements are sparse, represent an average SO2 column over a relatively large satellite pixel, while the Brewer observations are designed to provide a local point measurement".

Comment 9: "8.33-35. Thus the statement, "In all cases, however, the observed ... were always higher ..." is simply incorrect, as demonstrated with the numbers just above, and should be removed".

Answer to comment 9: The statement has been removed.

**Comment 10: "9.1-5. Why is there so much inconsistency between Figures 5 and 7. Fig. 7 shows OMI measurements of 1-4 DU across large regions of Europe, yet Fig. 5 indicates almost all OMI measurements < 1 DU and most measurements < 0.5 DU".**

**Answer to comment 10:** We would like to clarify that Figure 7 (now has become Fig. 6) does not show OMI measurements but forecasted calculations by the MACC model with and without OMI assimilation for 21 September 2014. On the other hand Fig. 5 (now has become Fig. 4) is based on actual measurements, in which OMI had only a couple of measurements over the Brewer sites.

**Comment 11: "Figure 9. The differences between the coloured lines are not obvious".**

Answer to comment 11: The figure has been redrawn to become clear.

**Comment 12: "10.15. What is meant by both methods?"**

**Answer to comment 12:** "It is clearly shown that the zero-calibrated Brewer  $SO_2$  data do not compare well with OMI and GOME-2 levels. Instead, the Langley calibrated Brewer data compare better with OMI and GOME-2 retrievals". This is clarified in the new text (see section 3.2, page 12, new lines 22-24).

**Comment 13: "Fig. 15. Why is the Brewer baseline at 0.2-0.3 DU for the stations under the plume, whereas for the 10 outside stations the baseline is closer to zero?"**

**Answer to comment 13:** It has to do with the offset of the instruments. We have overcome this problem by analysing departures from the non-perturbed ten days prior to the eruption as described before. The new Figure 12 (old figure 15) does not show this discrepancy anymore.

Comment 14: "11.38. Does an average  $SO_2$  plume of 0.1 DU mean anything when earlier the averages of the Brewers without influence by volcanoes was on the order of 0.4 DU? It does not help the authors' argument to be calling out numbers in the text which are in the noise of the measurements. The authors also never explain what a negative DU measurement means. What causes this? Are the negative numbers a real measurement?"

Answer to comment 14: No, it does not mean anything. All  $SO_2$  columns have been recalculated as departures from the non-perturbed 10-day baseline and we do not call out numbers which are in the noise of the measurements as can be seen in the new text (section 3.3).

With regard to the negative  $SO_2$  columns, we clarify in the revised section 2.1 that "From the above described operational Brewer algorithm it is evident that the estimation of columnar SO2 is the result of the difference between two columnar terms (O3 + SO2) and O3. Both terms have uncertainties (weighting functions, calibrations, random errors, systematic errors). Systematic negative values could be the result of a systematic offset in the measurements that can be related to the calibration of the instrument (usually optimized only for the ozone measurements). Randomly varying positive and negative values around zero, suggest that the signal of SO2 is small (and thus the difference of two terms should be close to zero) but since both terms have uncertainties, negative values are possible indicating that the amount of SO2 in the atmosphere is below the detection limit of the instrument and could be considered as noise. In this work we have repeated our analysis excluding the negative values and the results remained the same i.e. a positive increase after a major volcanic eruption was confirmed as described in the following sections".

**Comment 15: "Fig. 16. Why is a 7 day running mean now added to the measurements? Does it show something missing in the simple averaged daily data shown up to now?"**

**Answer to comment 15:** To avoid confusion the left panel of that figure has been removed. Please note that the new figure for Kasatochi is now Fig. 13 because the paragraph for Grimsvötn has been removed as requested by Reviewer #2.

Comment 16: "12.31-13.6. A calculation of Pearson's correlation coefficients is not necessary to convince the readers that the Brewers, GOME-2 and OMI are all in agreement at least over Europe. Is the Taiwan station included in the correlation coefficients? If so, does the fact that there is virtually no correlation there get masked because it is only one station? What is telling about this paragraph, and the corresponding Table 5, is that such tests were not used in any previous comparison, most certainly because the results would have been much worse, see Figures 5, 12, 15".

**Answer to comment 16:** No, Taiwan is not included in the correlation coefficients. Table 5 has been redrawn to show the correlation coefficients between the Brewers, GOME-2 and OMI over Europe in all three volcanic eruptions (Kasatochi, Eyjafjallajökull and Bárðarbunga). The correlations between the Brewers and GOME-2 were found to be statistically significant in all volcanic eruptions. Brewer and OMI data were strongly correlated in Kasatochi but unfortunately the sparsity of OMI data during Eyjafjallajökull and Bárðarbunga prevented us to calculate correlations between the Brewers and OMI during these two volcanoes, as described in the text.

**Comment 17: "13.16-18. This statement is based on only the Kasatochi results and does not hold for 4 of the 5 eruptions studied, thus the statement either has to be removed from the conclusions or dampened considerably by pointing out all the other times when no correlation or a poor correlation was found".**

**Answer to comment 17:** The statement has been removed and the new text now reads: "The Kasatochi eruption provided a formidable example for a volcanic SO2 plume to be observed not only by the ground based instruments, but from space-borne as well (OMI and GOME-2). Relative to the undisturbed period before Kasatochi the amplitude of the signal is 2 DU for GOME-2 and 1.5 DU for OMI. The results for the other volcanic eruptions are similar for the Brewer network, but unfortunately because of the sparsity of satellite overpassing the Brewer stations the satellite data concur with those from the Brewers only in Kasatochi".

**Reply to Reviewer #2**

The authors are indebted to Reviewer #2 for his valuable comments which have all been taken into account and appropriate revisions have been done as follows:

**Answers to main comments**

Comment 1: "The measurement capability of Brewer instruments should be better explained. Since the paper focuses on the detection of small  $SO_2$  signals, the methodology to derive  $SO_2$  total content should be summarized in the paper itself. An assessment of the mean  $SO_2$  values generally provided by Brewer instruments should be provided".

**Answer to comment 1:** The summary of the methodology to determine the  $SO_2$  column has been added in section 2.1. The requested assessment emerges from our answers to comments 2 and 3 below as well as in the literature by the papers of Fioletov et al. (1998, 2016) which are referred to in the text.

**Comment 2: "As optical instruments, the Brewer measurements can be perturbed by ash present in the volcanic plumes. This issue should be addressed in the article".**

**Answer to comment 2:** We have added a relevant comment in section 2.1, in which it is shown that the presence of volcanic ash is not expected to perturb the SO2 measurements, this addition reads as follows:

"Finally, we need to point out that perturbations by ash present in the volcanic plumes have been shown not to affect the Brewer  $SO_2$  measurements. This is based on the result of Pappalardo et al., 2013 paper based on EARLINET observations following the Eyjafjallajökull eruption in which they found that the Ångström exponent of the volcanic ash optical depth is close to zero. This indicates that the effect of ash in the UV and visible region on the aerosol extinction is almost independent from wavelength. The Brewer  $SO_2$  measurements taken in a narrow wavelength band in the UV are therefore not expected to be influenced by the presence of volcanic ash considering the weights already applied in the operational Brewer algorithm".

Pappalardo, G., Mona, L., D'Amico, G., et al.: Four-dimensional distribution of the 2010 Eyjafjallajökull volcanic cloud over Europe observed by EARLINET, Atmos. Chem. Phys., 13, 4429-4450, doi:10.5194/acp-13-4429-2013, 2013.

**Comment 3: "For readers not familiar with total $SO_2$ measurements by Brewer spectrometers, it is rather intriguing to see negative total $SO_2$ values. So it would be worth explaining why such negative values have to be considered in the general Brewer (and satellite) retrieval".**

**Answer to comment 3:** The following text has been added in section 2.1: "From the above described operational Brewer algorithm it is evident that the estimation of columnar SO2 is the result of the difference between two columnar terms (O3 + SO2) and O3. Both terms have uncertainties (weighting functions, calibrations, random errors, systematic errors). Systematic negative values could be the result of a systematic offset in the measurements that can be related to the calibration of the instrument (usually optimized only for the ozone measurements). Randomly varying positive and negative values around zero, suggest that

the signal of SO2 is small (and thus the difference of two terms should be close to zero) but since both terms have uncertainties, negative values are possible indicating that the amount of SO2 in the atmosphere is below the detection limit of the instrument and could be considered as noise. In this work we have repeated our analysis excluding the negative values and the results remained the same i.e. a positive increase after a major volcanic eruption was confirmed as described in the following sections".

**Comment 4: "Two lagrangian models are used for the analysis: FLEXPART and HYSPLIT. An explanation is needed on why two different models need to be used (paragraph 2.3)".**

**Answer to comment 4:** Both HYSPLIT and FLEXPART are well established modelling tools and both are widely used in relevant studies. As stated in the text we use FLEXPART-WRF for the dispersion simulations. FLEXPART-WRF is driven by WRF 1-hourly data at 45×45 km and the higher spatial and temporal resolution of meteorological fields allows a more detailed representation of the volcanic plume dispersion but have significant higher computational time. To overcome this computational cost problem we use HYSPLIT for the back-trajectories calculations. HYSPLIT is driven by lower temporal and spatial resolution meteorological fields, specifically with the GDAS 3-hourly meteorology at 1°×1° resolution (see revised paragraph 2.3).

**Comment 5: "In the case of the Bardarbunga volcano, the FLEXPART model has been used to simulate $SO_2$ levels in air masses sampled at Hohenpeissenberg station. But there is no detail on the simulation and on the initial emitted $SO_2$ levels".**

**Answer to comment 5:** We thank the reviewer for this notice. The following text is now added in section 3.1: "The simulation period is 18-26 September 2014. We assume a constant SO2 release rate of 119 kilotons per day as reported by Gíslason et al. (2015) from near the source SO2 measurements during the first weeks of the eruption. Similar emission rates are also suggested by Schmidt et al. (2015) through comparisons between NAME simulations (UK Met Office's Numerical Atmospheric-dispersion Modelling Environment) and OMI satellite retrievals. The emission height is set between 0 and 3500 m above ground level, consistent throughout the simulation period".

Schmidt, A., Leadbetter, S., Theys, N., Carboni, E., Witham, C. S., Stevenson, J. A., Birch, C. E., Thordarson, T., Turnock, S., Barsotti, S, Delaney, L., Feng, W., Grainger, R. G., Hort, M. C., Höskuldsson, A., Ialongo, I., Ilyinskaya, E., Jóhannsson, T., Kenny, P., Mather, T. A., Richards N. A. D., and Shepherd, J.: Satellite detection, long-range transport, and air quality impacts of volcanic sulfur dioxide from the 2014-2015 flood lava eruption at Bárðarbunga (Iceland), J. Geophys. Res. Atmos., 120, 9739-9757, doi:10.1002/2015JD023638, 2015.

Gíslason, S. R., Stefánsdóttir, G., Pfeffer, M. A., Barsotti, S., Jóhannsson, Th., Galeczka, I., Bali, E., Sigmarsson, O., Stefánsson, A., Keller, N. S., Sigurdsson, Á., Bergsson, B., Galle, B., Jacobo, V. C., Arellano, S., Aiuppa, A., Jónasdóttir, E. B., Eiríksdóttir, E. S., Jakobsson, S., Guðfinnsson, G. H., Halldórsson, S. A., Gunnarsson, H., Haddadi, B., Jónsdóttir, I., Thordarson, Th., Riishuus, M., Högnadóttir, Th., Dürig, T., Pedersen, G. B. M., Höskuldsson, Á., Gudmundsson, M.T.: Environmental pressure from the 2014-15 eruption of Bárðarbunga volcano, Iceland, Geochem. Persp. Let., 1, 84-93, 2015.

Comment 6: "For the same volcano, it is not completely clear that the elevated  $SO_2$  levels detected by ground stations correspond to the volcanic plume. Also a better explanation should be given on why the plume is not seen in OMI and GOME 2 measurements shown in Figure 5. The case for the detection of this volcanic plume by the satellite instruments over Europe and for the attribution of increased  $SO_2$  levels from these measurements (page 8) is not completely made".

**Answer to comment 6:** We would like to point out that the fact that the elevated SO2 levels detected by ground stations (Brewer network) corresponds to the volcanic SO2 plume was confirmed by performing the back trajectories analysis with the HYSPLIT dispersion model as well as from the FLEXPART and MACC model simulations. Additionally, the Reviewer #2 correctly points out that the plume is not seen in OMI and GOME-2 measurements shown in Figure 5 (new Figure 4). We would like to note that we have carefully revisited the OMI and GOME-2 data sets and found out that during the most perturbed period following the eruptions of Bárðarbunga (21-22 September 2014) the satellite overpasses were so sparse that the daily average was not corresponding to the Brewer network sample. For instance and following Bárðarbunga, there were many days where we had only one or two OMI measurements following the eruption, obviously not representing the 19 Brewer instruments in Europe. To temper our past conclusions we have applied a criterion according to which a daily average from either OMI or GOME-2 should be calculated if and only if more than half of the individual overpasses had data at each day. As can be seen from the revised figures 4, OMI results are missing for not meeting this criterion. Also GOME-2 results are missing from the figure during the peak period (21-22/9/2014) for not passing this criterion.

In spite of the sparsity of OMI observations post Bárðarbunga, it was thought that they could still be used as SO2 assimilated field in the SO2 analyses and forecasts produced with the MACC (Monitoring Atmospheric Composition and Climate) system (http://atmosphere.copernicus.eu/). This near-real-time forecasting system assimilates satellite observations to constrain modelling forecasts (Inness et al., 2015; Flemming et al., 2015). The OMI instrument on board the AURA satellite provided information about concentrations of volcanic SO2 emitted by the Icelandic Bárðarbunga volcano on 20 September; these observations were assimilated in 2014 by the MACC model in cases of volcanic eruptions, i.e. when OMI values exceeded 5 DU. As shown by the chart of total column SO2 obtained from http://atmosphere.copernicus.eu/ (Figure 6), the subsequent forecasts then captured the transport of this plume of volcanic SO2 southward spreading over the continent on 21 and 22 September. The plume stretched all the way from Finland through Poland, Germany and France, to southern England. A parallel forecast, for which no OMI data were used (Fig. 6, right), did not show any elevated SO2 values, confirming that 'normal' emissions of SO2 (including shipping and industrial activities) could not explain the observed situation. All the above are described in the revised text.

**Comment 7: "The fact that the 2011 Grimsvötn volcanic plume was not detected by the European Brewer instrument does not bring much to the article. This paragraph should be removed".**

**Answer to comment 7:** The paragraph for Grimsvötn and its figures have been removed.

Comment 8: "Again for the Eyjafjallajökull volcano, OMI and GOME 2 do not seem to detect the  $SO_2$  signal. An explanation is needed on the lack of detection by satellite instruments. Also, the left panel of Figure 16 is redundant with the right panel".

**Answer to comment 8:** For the case of Eyjafjallajökull, OMI and GOME-2 do not seem to detect the SO2 signal because the satellite data were sparse, particularly OMI.

To avoid confusion the left panel of Fig. 16 has been removed. Please note that the new figure for Kasatochi is now Fig. 13.

Comment 9: "2008 Kasatochi case: it is not clear from the article why the plume is not detected in Taiwan by the satellite instruments, contrary to the observations in Europe and North America. This issue should be addressed".

**Answer to comment 9:** During the revision of the manuscript we analysed back trajectories from Taiwan for the days of elevated  $SO_2$  observed by the Brewer, something that has been overlooked in the first version of the paper. The analysis showed that the air masses did not originate from Kasatochi. To avoid confusion we have removed Taiwan from the figure of Kasatochi (see new Figure 13).

Comment 10: "The conclusion should better summarize in which general conditions (SO2 levels, time after eruption) Brewer instruments can be useful for the detection of SO2 volcanic plumes. The article is qualitative in general and such a summary would provide a quantified assessment of the measurements capability of Brewer instruments with respect to SO2 measurements. Comparison with OMI and GOME 2 measurements capacity in similar cases would be useful. It would be also worth mentioning why IASI and AIRS measurements are not included in the analysis".

Answer to comment 10: The concluding section has been fully revised in the new manuscript taking into consideration all the above useful comments. The second paragraph in the Conclusion has been revised and reads as follows: "From the results discussed in section 3 some general remarks can be put forward concerning SO2 levels and detection time after the eruption. Starting with the Kasatochi eruption, as it appears from Figure 13, the plume can be detected 4 days after the eruption over Canada and the US and about 7 days over Europe with an average amplitude on the order of 2 DU compared to the unperturbed ten day pre-volcanic period (baseline). All estimates are based obviously on measurements taken under the plume. The Kasatochi eruption provided a formidable example for a volcanic  $SO_2$  plume to be observed not only by the ground based instruments, but from space-borne as well (OMI and GOME-2). Relative to the undisturbed period before Kasatochi the amplitude of the signal is 2 DU for GOME-2 and 1.5 DU for OMI. The results for the other volcanic eruptions are similar for the Brewer network, but unfortunately because of the sparsity of satellite overpassing the Brewer stations the satellite data concur with those from the Brewers only in Kasatochi. Based on the above discussion it appears that currently no single network can independently and fully monitor the evolution of volcanic SO2 plumes. Among a few reasons are lack of measurements during peak values, complications from meteorological events, ejection heights and exposure conditions. The evidence presented here points that combination of observations from various instruments, aided by chemical transport models and operated in synergy could address such a complex issue".

Additionally, we want to point out that we did not consider in this paper  $SO_2$  measurements from IASI and AIRS since both instruments are IR spectroradiometers, while OMI and GOME-2 data are based on UVB/Vis spectroradiometers whose retrieval algorithms rely on the differential optical absorption in the UV band which is also the case with the Brewer instrument. A Brewer-IASI or Brewer-AIRS comparison would also have to consider differences in the spectroscopy and algorithm concept and thus would require further analysis which is beyond the scope of this paper.

**Answers to minor comments**

Comment 11: "In general, figures' legends should be more informative, with the description of the various plots and the name of the volcano case to which the figure refer (when  $SO_2$  levels are plotted)".

**Answer to 11:** The figures' legends have been re-written to be more informative as suggested by the reviewer.

**Comment 12: "Figure 7: can the authors comment on the spot of elevated $SO_2$ observed between Italy and Greece?"**

**Answer to 12:** The spot of elevated  $SO_2$  between Italy and Greece is related to the Etna volcano and is a result of using continuous natural  $SO_2$  emissions that might be too high in the MACC model.

**Further additions to the manuscript**

Three more stations have been added, namely Regina and Goose Bay in Canada and Mauna Loa in the US. Two more co-authors have been added, Vitali Fioletov and Irina Petropavlovskikh, who provided the SO2 column data for these additional stations.

**Detecting volcanic sulfur dioxide plumes in the Northern Hemisphere using the Brewer spectrophotometers, other networks, and satellite observations**

Christos S. Zerefos1,2,3,4, Kostas Eleftheratos2,5, John Kapsomenakis1, Stavros Solomos6, Antje Inness7, Dimitris Balis8, Alberto Redondas9, Henk Eskes10, Marc Allaart10, Vassilis Amiridis6, 5 Arne Dahlback11, Veerle De Bock12, Henri Diémoz13, Ronny Engelmann14, Paul Eriksen15, Vitali Fioletov16, Julian Gröbner17, Anu Heikkilä18, Irina Petropavlovskikh19, Janusz Jarosławski20, Weine Josefsson21, Tomi Karppinen22, Ulf Köhler23, Charoula Meleti8, Christos Repapis4, John Rimmer24, Vladimir Savinykh25, Vadim Shirotov26, Anna Maria Siani27, Andrew R. D. Smedley24, Martin Stanek28, René Stübi29

10

1Research Centre for Atmospheric Physics and Climatology, Academy of Athens, Athens, Greece 2Biomedical Research Foundation, Academy of Athens, Athens, Greece Navarino Environmental Observatory (N.E.O.), Messinia, Greece 4Mariolopoulos-Kanaginis Foundation for the Environmental Sciences, Athens, Greece 5Faculty of Geology and Geoenvironment, National and Kapodistrian University of Athens, Greece

15 6Institute for Astronomy, Astrophysics, Space Applications and Remote Sensing (IAASARS), National Observatory of Athens, Athens, Greece 7European Centre for Medium-Range Weather Forecasts (ECMWF), Reading, UK

- 8Department of Physics, Aristotle University of Thessaloniki, Thessaloniki, Greece 20 9Izaña Atmospheric Research Center, AEMET, Tenerife, Canary Islands, Spain 10Royal Netherlands Meteorological Institute (KNMI), De Bilt, the Netherlands 11Department of Physics, University of Oslo, Oslo, Norway 12Royal Meteorological Institute of Belgium, Brussels, Belgium 13ARPA Valle d'Aosta, Saint-Christophe, Italy
- 14Leibniz Institute for Tropospheric Research, Leibniz, Germany 25 15Danish Meteorological Institute, Copenhagen, Denmark 16Environment and Climate Change Canada, Toronto, Canada 17PMOD/WRC, Davos Dorf, Switzerland
- 18Climate Change Unit, Finnish Meteorological Institute, Helsinki, Finland 19Cooperative Institute for Research in Environmental Sciences, University of Colorado, Boulder, CO USA 30 20Institute of Geophysics, Polish Academy of Sciences, Warsaw, Poland 21Swedish Meteorological and Hydrological Institute, Norrköping, Sweden 22Arctic Research Centre, Finnish Meteorological Institute, Sodankylä, Finland

 23DWD, Meteorological Observatory Hohenpeissenberg, Germany
 24Centre for Atmospheric Science, School of Earth, Atmospheric and Environmental Sciences, University of 35 Manchester, Manchester M13 9PL, UK A.M. Obukhov Institute of Atmospheric Physics, Kislovodsk, Russia 26Institute of Experimental Meteorology, Obninsk, Russia

27Department of Physics, Sapienza – University of Rome, Rome, Italy

Christos S. Zerefos1,2,3,10, Kostas Eleftheratos2,4, John Kapsomenakis1, Stavros Solomos5, Antje Inness6, Dimitris Balis7, Alberto Redondas8, Henk Eskes9, Vassilis Amiridis5, Christos Repapis40, Marc Allaart9, Ronny Engelmann44, Arne Dahlback42, Veerle De Bock43, Henri Diémoz44, Paul Eriksen45, Julian Gröbner46, Anu Heikkilä47, Janusz Jarosławski48, Weine Josefsson49, Tomi Karppinen20, Ulf Köhler24, Charoula Meleti7, John Rimmer22, Vladimir Savinykh23, Vadim Shirotov24, Anna Maria Siani25, Andrew R. D. Smedley22, Martin Stanek26, René Stübi27 45

<sup>28Solar and Ozone Observatory, Czech Hydrometeorological Institute, Hradec Kralove, Czech Republic 40 29Federal Office of Meteorology and Climatology, MeteoSwiss, Payerne, Switzerland

<sup>1Research Centre for Atmospheric Physics and Climatology, Academy of Athens, Athens, Greece

2Biomedical Research Foundation, Academy of Athens, Athens, Greece 3Navarino Environmental Observatory (N.E.O.), Messinia, Greece 4Faculty of Geology and Geoenvironment, University of Athens, Greece 5Institute for Astronomy, Astrophysics, Space Applications and Remote Sensing (IAASARS), National 5 **Observatory of Athens, Athens, Greece** 6European Centre for Medium Range Weather Forecasts (ECMWF), Reading, UK 2Department of Physics, Aristotle University of Thessaloniki, Thessaloniki, Greece 8Izaña Atmospheric Research Center, AEMET, Tenerife, Canary Islands, Spain 9Royal Netherlands Meteorological Institute (KNMI), De Bilt, the Netherlands 40Mariolopoulos Kanaginis Foundation for the Environmental Sciences, Athens, Greece 10 44Leibniz Institute for Tropospheric Research, Leibniz, Germany 42Department of Physics, University of Oslo, Oslo, Norway 13Royal Meteorological Institute of Belgium, Brussels, Belgium 44ARPA Valle d'Aosta, Saint Christophe, Italy 15Danish Meteorological Institute, Copenhagen, Denmark 15 16PMOD/WRC, Davos Dorf, Switzerland 47Climate Change Unit, Finnish Meteorological Institute, Helsinki, Finland 48Institute of Geophysics, Polish Academy of Sciences, Warsaw, Poland 49Swedish Meteorological and Hydrological Institute, Norrköping, Sweden 20Arctic Research Centre, Finnish Meteorological Institute, Sodankylä, Finland 20 21DWD, Meteorological Observatory Hohenpeissenberg, Germany 22Centre 
[revised manuscript text omitted]

|                    | Latitude          | Longitude          | Elevation asl (m) | Instruments      | Data source       |
|--------------------|-------------------|--------------------|-------------------|------------------|-------------------|
| SODANKYLA   | <del>67.36</del>  | <del>26.63</del>   | <del>180</del>    | Brewer MKII 037  | FMI               |
| VINDELN            | <del>64.24</del>  | <del>19.77</del>   | <del>225</del>    | Brewer MKII 006  | SMHI       |
| JOKIOINEN          | <del>60.82</del>  | <del>23.50</del>   | <del>106</del>    | Brewer MKIII 107 | FMI               |
| <del>OSLO</del>    | <del>59.90</del>  | <del>10.73</del>   | <del>50</del>     | Brewer MKV 042   | U_Oslo            |
| CHURCHILL          | <del>58.74</del>  | <del>-93.82</del>  | <del>16</del>     | Brewer MKII 026, | WOUDC      |
|                    |                   |                    |                   | Brewer MKIV 032, |                   |
|                    |                   |                    |                   | Brewer MKIII 203 |                   |
| NORRKOEPING | <del>58.58</del>  | <del>16.15</del>   | <del>43</del>     | Brewer MKIII 128 | SMHI       |
| COPENHAGEN  | <del>55.63</del>  | <del>12.67</del>   | <del>50</del>     | Brewer MKIVe 082 | <del>DMI</del>    |
| <del>OBNINSK</del> | <del>55.10</del>  | <del>36.60</del>   | <del>100</del>    | Brewer MKII 044  | IEM-SPA           |
| EDMONTON    | <del>53.55</del>  | <del>-114.10</del> | <del>766</del>    | Brewer MKII 055, | WOUDC      |
|                    |                   |                    |                   | Brewer MKIV 022  |                   |
| MANCHESTER         | <del>53.47</del>  | -2.23              | <del>76</del>     | Brewer MKIII 172 | U_Manchester      |
| WARSAW             | <del>52.17</del>  | <del>20.97</del>   | <del>107</del>    | Brewer MKIII 207 | PAS-IGF           |
| DE BILT            | <del>52.10</del>  | <del>5.18</del>    | 2                 | Brewer MKIII 189 | KNMI              |
| BELSK              | <del>51.84</del>  | <del>20.79</del>   | <del>180</del>    | Brewer MKII 064  | PAS-IGF           |
| READING            | <del>51.44</del>  | <del>-0.94</del>   | <del>66</del>     | Brewer MKIV 075, | U_Manchester      |
|                    |                   |                    |                   | Brewer MKII 126  |                   |
| UCCLE              | <del>50.80</del>  | 4 <del>.36</del>   | <del>100</del>    | Brewer MKII 016, | RMIB              |
|                    |                   |                    |                   | Brewer MKIII 178 |                   |
| HRADEC KRALOVE     | <del>50.18</del>  | <del>15.84</del>   | <del>285</del>    | Brewer MKIII 184 | CHMI-HK           |
| SATURNA ISLAND     | <del>48.78</del>  | <del>-123.13</del> | <del>178</del>    | Brewer MKII 012  | WOUDC             |
| HOHENPEISSENBERG   | 4 <del>7.80</del> | <del>11.01</del>   | <del>985</del>    | Brewer MKII 010  | DWD-MOHp          |
| <del>DAVOS</del>   | <del>46.81</del>  | <del>9.84</del>    | <del>1590</del>   | Brewer MKIII 163 | PMOD/WRC          |
| AROSA              | <del>46.78</del>  | <del>9.67</del>    | <del>1840</del>   | Brewer MKII 040, | MeteoSwiss |
|                    |                   |                    |                   | Brewer MKIII 156 |                   |
| AOSTA              | 4 <del>5.74</del> | <del>7.36</del>    | <del>569</del>    | Brewer MKIV 066  | ARPA-VDA          |
| TORONTO            | 4 <del>3.78</del> | <del>-79.47</del>  | <del>198</del>    | Brewer MKII 015  | WOUDC             |
| KISLOVODSK         | 4 <del>3.73</del> | 4 <del>2.66</del>  | <del>2070</del>   | Brewer MKII 043  | RAS-IAP           |
| ROME               | 4 <del>1.90</del> | <del>12.52</del>   | <del>75</del>     | Brewer MKIV 067  | U_Rome            |
| THESSALONIKI       | 4 <del>0.63</del> | <del>22.95</del>   | <del>60</del>     | Brewer MKII 005  | AUTH              |
| NIWOT RIDGE        | 40.03             | -105.53            | <del>2891</del>   | Brewer MKIV 146  | NEUBrew           |
| ATHENS             | <del>37.99</del>  | <del>23.78</del>   | <del>191</del>    | Brewer MKIV 001  | BRFAA             |

**Table 3. Rural AirBase stations analysed in this study (see text).**

| Station ID | Station name                 | Latitude | Longitude | Closest Brewer (within 150 km) |
|------------|------------------------------|----------|-----------|--------------------------------|
| GB0583A    | Middlesbrouth                | 54.569   | -1.221    | Manchester                     |
| NL00444    | De Zilk-Vogelaarsdreef       | 52.298   | 4.51      | Uccle                          |
| PL0105A    | Parzniewice                  | 51.291   | 19.517    | Belsk                          |
| NL00133    | Wijnandsrade-Opfergeltstraat | 50.903   | 5.882     | De Bilt                        |
| GB0038R    | Lullington Heath             | 50.794   | 0.181     | Reading                        |
| CH0005R    | Rigi                         | 47.067   | 8.463     | Arosa                          |
| CH0002R    | Payerne                      | 46.813   | 6.944     | Aosta                          |

| Table 4. SO 2 | column departures a | t mid-latitude stations | averaged in bimonthly | periods following | volcanic | eruptions. |
|--------------------------|---------------------|-------------------------|-----------------------|-------------------|----------|------------|
|                          |                     |                         |                       |                   |          |            |

|                  |              | August-September 2008
(Kasatochi) |            | April-M
(Eyjafjal | ay 2010
lajökull) | September-October 2014
(Bárðarbunga) |            |
|------------------|--------------|--------------------------------------|------------|-----------------------------|-----------------------------|-----------------------------------------|------------|
| (a)       | Latitude     | mean                                 |     | mean                        |                      | mean                                    | σ   |
| Sodankÿla | 67.36 | 0.6                           | 2.1 | 0.1                  | 0.7                  | -0.5                             | 1.8 |
| Vindeln          | 64.24 | 0.4                           | 1.4 | 0.0                  | 0.4                  | -0.2                             | 0.9 |

| Jokioinen               | 60.82 | 0.5                                     | 0.6         | *                                        | *                       | 0.4                                      | 0.5             |
|-------------------------|--------------|-----------------------------------------|--------------------|------------------------------------------|-------------------------|------------------------------------------|-----------------|
| Oslo                    | 59.90        | *                                       | *                  | 0.7                                      | 0.6                     | -0.1                                     | 1.0             |
| Churchill               | 58.74 | 0.6                              | 0.8         | -0.3                              | 1.1              | 0.4                               | 1.0      |
| Norrkoeping      | 58.58 | 0.4                              | 0.8         | -0.1                              | 0.2              | 0.1                               | 0.8      |
| Copenhagen              | 55.63 | 0.3                              | 0.8         | 0.5                               | 0.9              | -0.4                              | 0.7      |
| Obninsk          | 55.10 | *                                       | *                  | 0.1                               | 0.5              | 0.3                               | 0.9      |
| Edmonton                | 53.55 | 0.4                              | 0.6         | 0.4                               | 0.4              | 0.0                               | 0.4      |
| Manchester              | 53.47 | 0.6                              | 0.7         | 0.0                               | 0.6              | 0.4                               | 1.6      |
| Goose Bay               | 53.29 | 0.2                              | 0.4         | *                                        | *                       | 0.3                               | 0.3      |
| Warsaw                  | 52.17 | *                                       | *                  | *                                        | *                       | 0.1                               | 0.4      |
| De Bilt          | 52.10 | 0.1                              | 0.9         | -0.3                              | 0.9              | 0.2                               | 0.8      |
| Belsk            | 51.84 | 0.3                              | 0.6         | -0.4                              | 0.4              | 0.4                               | 0.5      |
| Reading          | 51.44 | 0.2                              | 0.7         | 1.2                               | 1.2              | 0.3                               | 1.7      |
| Uccle            | 50.80 | 0.1                              | 0.6         | -0.5                              | 0.6              | 0.7                               | 1.3      |
| Regina           | 50.20 | 0.0                              | 0.9         | *                                        | *                       | *                                        | *               |
| Hradec Kralove          | 50.18 | 0.2                              | 0.4         | -0.3                              | 0.4              | -0.6                              | 0.7      |
| Saturna Island   | 48.78 | 0.4                              | 1.1         | 0.0                               | 0.2              | 0.4                               | 0.5      |
| Hohenpeissenberg | 47.80 | 0.0                              | 0.5         | 0.5                               | 0.6              | -0.1                              | 1.6      |
| Davos            | 46.81 | 0.2                              | 0.5         | -0.1                              | 0.3              | -0.1                              | 0.2      |
| Arosa                   | 46.78 | 0.6                              | 1.5         | -0.5                              | 1.5              | -0.1                              | 0.5      |
| Aosta            | 45.74 | -0.1                             | 0.6         | 0.0                               | 0.6              | -0.6                              | 0.8      |
| Toronto          | 43.78 | 0.5                              | 1.0         | -0.2                              | 0.5              | 0.4                               | 0.5      |
| Kislovodsk       | 43.73 | -0.3                             | 0.3         | -0.1                              | 0.3              | 0.2                               | 0.2      |
| Rome                    | 41.90 | -0.1                             | 1.1         | -0.8                              | 1.3              | -0.2                              | 0.5      |
| Thessaloniki     | 40.63 | 0.4                              | 0.7         | -0.7                              | 0.9              | *                                        | *               |
| Boulder                 | 40.03 | 0.1                              | 0.5         | 0.1                               | 0.9              | *                                        | *               |
| Athens                  | 37.99 | 0.9                              | 0.8         | -0.4                              | 0.6              | 0.0                               | 0.4      |
| (b)              |              | $\underline{\text{mean} \pm \text{st}}$ | . error (N) | $\underline{\text{mean} \pm \text{st.}}$ | error (N)               | $\underline{\text{mean} \pm \text{st.}}$ | error (N)       |
| All Brewers             |              | $0.29 \pm 0.0$                          | 03 (1051)   | $-0.04 \pm 0.1$                          | $-0.04 \pm 0.03$ (1064) |                                          | 03 (861) |
| GOME-2                  |              | $0.23 \pm 0.02 (1057)$                  |                    | $0.08 \pm 0.01$ (971)                    |                         | $-0.03 \pm 0.02$ (677)                   |                 |
| OMI (TRM)        |              | $0.15 \pm 0.015$                        | 02 (741)    | $0.00 \pm 0.00$                          | 02 (438)         | $0.01 \pm 0.01$                          | 02 (395) |

(\*) missing values are those possessing < 25 days of data in each bimonthly period, or no data.

Table 5. Correlation coefficients between the mean columnar SO2 measured by the brewers in Europe and provided by the satellite products of OMI and GOME-2 during the volcanic eruptions of Kasatochi (2008), Eyjafjallajökull (2011) and Bárðarbunga (2014) for stations located under the volcanic SO2 plume.

| Europe                                                                                                             | August-September 2008              | April-May 2010         | September-October 2014 |  |  |  |  |  |  |
|---------------------------------------------------------------------------------------------------------------------------|------------------------------------|-------------------------------|------------------------|--|--|--|--|--|--|
| Brewers and GOME-2                                                                                                        | 0.86 [59] (p<0.0001)               | 0.31 [54] (p=0.02336)  | 0.44 [39] (p=0.00496)  |  |  |  |  |  |  |
| Brewers and OMI (TRM)                                                                                                     | 0.86 [50] (p<0.0001)        | (*) [16]               | (*) [16]        |  |  |  |  |  |  |
| GOME-2 and OMI (TRM)                                                                                                      | 0.92 [48] (p<0.0001)               | (*) [15]               | (*) [15]        |  |  |  |  |  |  |
| Bold: all the above correlation                                                                                           | s are significant at confidence le | evel 95% or greater (t-test). |                        |  |  |  |  |  |  |
| (*): missing correlations are those possessing less than 30 days of data in each bimonthly period. In brackets: number of |                                    |                               |                        |  |  |  |  |  |  |
| pairs.                                                                                                                    |                                    |                               |                        |  |  |  |  |  |  |

10

Table 4. SO2 columns at mid-latitude stations averaged in bimonthly periods which include volcanic eruptions.

|                      |                   | Augu            | st-Septem             | ber 2008            | A                | April-May 2010              |                     | May-June 2011               |                   |                             | September-October 2014 |                |                     |
|----------------------|-------------------|-----------------|-----------------------|---------------------|------------------|-----------------------------|---------------------|-----------------------------|-------------------|-----------------------------|------------------------|----------------|---------------------|
| <del>(a)</del>       | Latitude          | mean            | ф.                    | <del>N (days)</del> | mean             | ਂ ਚ                         | <del>N (days)</del> | mean                        | ,
Ф            | <del>N (days)</del>         | mean                   | σ              | <del>N (days)</del> |
| SODANKYLA            | <del>67.36</del>  | <del>0.7</del>  | <del>1.9</del>        | 41                  | <del>-0.5</del>  | <del>0.6</del>              | 44                  | 0.1                         | <del>0.6</del>    | <del>59</del>               | <del>0.7</del>         | <del>1.8</del> | 27                  |
| VINDELN              | <del>64.24</del>  | <del>0.5</del>  | <del>1.2</del>        | <del>45</del>       | <del>0.4</del>   | <del>0.4</del>              | <del>49</del>       | [-3.2]                      | <del>0.8</del>    | <del>56</del>               | <del>0.3</del>         | <del>0.8</del> | <del>33</del>       |
| <del>JOKIOINEN</del> | <del>60.82</del>  | <del>0.4</del>  | <del>0.6</del>        | <del>42</del>       | *         | *                    | *            | 0.2                         | <del>0.3</del>    | <del>53</del>               | <del>0.6</del>         | <del>0.5</del> | <del>30</del>       |
| <del>OSLO</del>      | <del>59.90</del>  | *        | *              | *            | -1.7             | <del>0.7</del>              | <del>52</del>       | <del>0.9</del>              | <del>0.8</del>    | <del>51</del>               | <del>-0.1</del>        | <del>0.9</del> | 41                  |
| CHURCHILL            | <del>58.74</del>  | <del>0.6</del>  | <del>0.9</del>        | <del>42</del>       | 1.5              | <del>1.1</del>              | <del>47</del>       | 2.2                         | <del>0.8</del>    | <del>45</del>               | <del>0.3</del>         | <del>0.9</del> | <del>25</del>       |
| NORRKOEPING          | <del>58.58</del>  | <del>0.2</del>  | <del>0.8</del>        | 41                  | 0.0              | <del>0.2</del>              | <del>50</del>       | <del>0.7</del>              | <del>0.3</del>    | <del>59</del>               | <del>0.3</del>         | <del>0.7</del> | <del>39</del>       |
| COPENHAGEN    | <del>55.63</del>  | <del>1.6</del>  | <del>0.8</del>        | <del>55</del>       | -0.4             | <del>0.9</del>              | 4 <del>8</del>      | <del>0.7</del>              | <del>0.8</del>    | <del>31</del>               | <del>2.6</del>         | <del>0.6</del> | <del>38</del>       |
| <del>OBNINSK</del>   | <del>55.10</del>  | *        | *              | *            | <del>0.3</del>   | <del>0.6</del>              | <del>57</del>       | <del>0.6</del>              | <del>0.4</del>    | <del>58</del>               | <del>-0.1</del>        | <del>0.9</del> | <del>40</del>       |
| EDMONTON      | <del>53.55</del>  | <del>-0.2</del> | <del>0.5</del>        | <del>56</del>       | -1.0             | <del>0.5</del>              | <del>53</del>       | 1.5                         | <del>1.1</del>    | <del>56</del>               | *               | *       | <del>12</del>       |
| MANCHESTER    | <del>53.47</del>  | <del>0.6</del>  | <del>0.7</del>        | <del>35</del>       | <del>0.7</del>   | <del>0.6</del>              | <del>46</del>       | <del>0.9</del>              | <del>0.5</del>    | <del>40</del>               | <del>0.1</del>         | <del>1.5</del> | <del>31</del>       |
| WARSAW               | <del>52.17</del>  | *        | *              | *            | *         | *                    | *            | *                    | *          | *                    | <del>0.9</del>         | <del>0.4</del> | <del>45</del>       |
| DEBILT               | <del>52.10</del>  | <del>0.5</del>  | <del>0.8</del>        | <del>61</del>       | <del>0.4</del>   | <del>0.9</del>              | <del>61</del>       | <del>0.0</del>              | <del>0.6</del>    | <del>61</del>               | <del>0.3</del>         | <del>0.8</del> | <del>53</del>       |
| BELSK                | <del>51.84</del>  | <del>1.0</del>  | <del>0.5</del>        | <del>46</del>       | <del>1.1</del>   | <del>0.4</del>              | <del>45</del>       | <del>0.9</del>              | <del>0.4</del>    | <del>47</del>               | <del>0.6</del>         | <del>0.5</del> | <del>50</del>       |
| READING       | <del>51.44</del>  | <del>-0.3</del> | <del>0.7</del>        | <del>36</del>       | -1.4             | <del>1.5</del>              | <del>57</del>       | <del>1.1</del>              | <del>0.6</del>    | 4 <del>9</del>              | <del>-0.1</del>        | <del>1.5</del> | 4 <del>5</del>      |
| UCCLE                | <del>50.80</del>  | <del>0.7</del>  | <del>0.5</del>        | 4 <del>6</del>      | <del>-0.3</del>  | <del>0.6</del>              | <del>50</del>       | <del>-0.3</del>             | <del>0.5</del>    | <del>54</del>               | <del>1.6</del>         | <del>1.2</del> | 4 <del>3</del>      |
| HRADEC KRALOVE       | <del>50.18</del>  | <del>0.5</del>  | <del>0.4</del>        | <del>47</del>       | <del>0.3</del>   | <del>0.4</del>              | 44                  | <del>0.4</del>              | <del>0.4</del>    | <del>52</del>               | <del>0.6</del>         | <del>0.8</del> | <del>42</del>       |
| SATURNA ISLAND       | 4 <del>8.78</del> | <del>-0.4</del> | <del>1.2</del>        | <del>53</del>       | <del>-0.3</del>  | <del>0.2</del>              | <del>55</del>       | <del>1.4</del>              | <del>0.4</del>    | <del>54</del>               | <del>0.6</del>         | <del>0.5</del> | 4 <del>5</del>      |
| HOHENPEISSENBERG     | <del>47.80</del>  | <del>-0.1</del> | <del>0.5</del>        | <del>52</del>       | <del>0.4</del>   | <del>0.6</del>              | <del>48</del>       | <del>0.5</del>              | <del>0.5</del>    | <del>42</del>               | <del>0.8</del>         | <del>1.4</del> | <del>52</del>       |
| <del>DAVOS</del>     | 4 <del>6.81</del> | <del>0.5</del>  | <del>0.5</del>        | 4 <del>2</del>      | <del>0.6</del>   | <del>0.3</del>              | 4 <del>2</del>      | *                    | *          | <del>15</del>               | <del>2.0</del>         | <del>0.2</del> | <del>55</del>       |
| AROSA                | 4 <del>6.78</del> | <del>0.5</del>  | <del>1.4</del>        | <del>61</del>       | <del>1.3</del>   | <del>1.8</del>              | <del>59</del>       | 1.7                         | <del>1.2</del>    | <del>61</del>               | <del>-0.3</del>        | <del>0.6</del> | <del>59</del>       |
| AOSTA                | <del>45.74</del>  | <del>0.2</del>  | <del>0.5</del>        | <del>53</del>       | <del>0.0</del>   | <del>0.6</del>              | <del>52</del>       | <del>0.2</del>              | <del>0.5</del>    | <del>29</del>               | <del>1.1</del>         | <del>0.8</del> | <del>43</del>       |
| TORONTO              | 4 <del>3.78</del> | <del>-0.3</del> | <del>0.9</del>        | 4 <del>9</del>      | <del>-0.6</del>  | <del>0.5</del>              | <del>52</del>       | <del>0.7</del>              | <del>1.2</del>    | <del>33</del>               | <del>1.8</del>         | <del>0.5</del> | <del>39</del>       |
| KISLOVODSK           | <del>43.73</del>  | <del>-0.2</del> | <del>0.3</del>        | <del>40</del>       | <del>0.3</del>   | <del>0.2</del>              | <del>49</del>       | <del>0.3</del>              | <del>0.4</del>    | 44                          | <del>0.1</del>         | <del>0.2</del> | <del>50</del>       |
| ROME                 | 4 <del>1.90</del> | <del>1.2</del>  | <del>1.0</del>        | <del>57</del>       | <del>[4.4]</del> | <del>1.2</del>              | <del>50</del>       | <del>[4.6]</del>            | <del>0.6</del>    | <del>58</del>               | <del>0.5</del>         | <del>0.5</del> | <del>56</del>       |
| THESSALONIKI         | 4 <del>0.63</del> | <del>0.4</del>  | <del>0.7</del>        | <del>54</del>       | <del>0.9</del>   | <del>0.9</del>              | <del>49</del>       | <del>1.9</del>              | $\frac{1.0}{1.0}$ | <del>53</del>               | *               | *       | *            |
| NIWOT RIDGE          | <del>40.03</del>  | <del>-0.4</del> | <del>0.5</del>        | <del>56</del>       | <del>-1.1</del>  | <del>0.9</del>              | <del>45</del>       | <del>-0.7</del>             | <del>0.4</del>    | <del>54</del>               | *               | *       | *            |
| ATHENS               | <del>37.99</del>  | <del>1.6</del>  | <del>0.8</del>        | <del>55</del>       | <del>0.4</del>   | <del>0.7</del>              | <del>53</del>       | <del>[4.3]</del>            | <del>1.4</del>    | <del>53</del>               | <del>0.9</del>         | <del>0.4</del> | 44                  |
| <del>(b)</del>       |                   | Ħ               | <del>nean ± st.</del> | error               | m                | <del>mean ± st. error</del> |                     | <del>mean ± st. error</del> |                   | <del>mean ± st. error</del> |                        |                |                     |
| All Brewers          |                   |                 | $0.41 \pm 0.1$        | <del>12</del>       |                  | $0.05 \pm 0.$               | <del>12</del>       |                             | $0.72 \pm 0.15$   |                             |                        | $0.67 \pm 0.2$ | 15                  |
| GOME-2               |                   |                 | $0.26 \pm 0.26$       | <del>02</del>       |                  | $\frac{0.01\pm0.}{}$        | <del>01</del>       |                             | $0.02 \pm 0.00$   | <del>01</del>               |                        | $0.08 \pm 0.$  | <del>02</del>       |
| OMI (TRM)            |                   |                 | $0.04 \pm 0.01$       | <del>02</del>       |                  | $0.03 \pm 0.03$             | <del>02</del>       |                             | $0.03 \pm 0.03$   | <del>.02</del>              |                        | $-0.06 \pm 0$  | <del>.02</del>      |

(\*) missing values are those possessing

---

## Author Response (AR2)

**Reply to Reviewer #1**

*Minor comment 1: "1.47-48. Fix this awkward sentence as follows, …significant columnar SO2 increases, exceeding on the average 2 DU relative to an unperturbed pre-volcanic 10-day baseline, with a mean close to zero and σ = 0.46, as calculated..."*

**Answer to 1:** The sentence has been modified as suggested.

*Minor comment 2: "2.22-23. Suggest … quantities of both ash and SO2 are released which in the case of ash can result in air travel disruptions…"*

**Answer to 2:** The sentence has been modified as suggested.

*Minor comment 3: "7.39-8.5. The authors need to explain what the ground based measurements are in this statement, "… the columnar SO2 measurements over Europe were also compared with measurements from ground level European stations from the European Environment Agency databases (AirBase) covering…". I have no idea what "measurement" the columnar so2 measurements are being compared to."*

**Answer to 3:** The columnar SO2 measurements are compared to surface SO2 measurements. The text has been revised and it now reads as follows: "Only for the case of the Bárðarbunga eruption in 2014, the columnar SO2 measurements over Europe were also compared with surface SO2 measurements from ground based European stations. The surface SO2 data were obtained from the European Environment Agency databases (AirBase; http://www.eea.europa.eu/data-and-maps/data/aqereporting-1#tab-european-data) covering the bimonthly period September-October 2014."

*Minor comment 4: "8.21-35. Authors should explain why the STL product is not used, since it seems that would be the correct altitude for explosive volcanic eruptions versus volcanic degassing for TRM. I was surprised when STL was not mentioned further after the explanation of what it was."*

**Answer to 4:** The TRM retrievals are recommended by Ialongo et al. (2015) to be used for volcanic degassing at all altitudes. On the other hand, STL data are intended for use with explosive volcanic eruptions where the volcanic cloud is placed in the upper troposphere / stratosphere (UTLS). The reviewer correctly points this out, and in the revised manuscript we show the results from both TRM and STL products. The STL calculations have been added to the figures and in the text, and the results are almost identical as described in the revised text.

***Minor comment 5:*** *"Section 3.1 The 2014 Bárðarbunga case – What is difficult to understand about this case is the timing of the ejected material from the volcano. The dispersion modeling is from 18-26 Sept. The back trajectories shown are 5 days from about 20 Sept., so a similar period. In contrast the eruption is indicated to occur at day 0 in figures 4 and 5 with the plume not appearing until 20 days later. How does the reader reconcile this significant time difference? Did the outgassed so2 hang around Iceland for 15 days prior to being dispersed? I doubt it. Was the volcano continuing to outgas for a number of weeks? The authors need to be clear and not mark the eruption as a one day event if that is not the case. If that is the case then the authors need to explain why the so2 was located over Iceland 15 days later."*

**Answer to 5:** Bárðarbunga was continuously active during September – October 2014 but it was only during 18-26 September when meteorological conditions favoured transport towards Europe as shown by back trajectories analyses (see Figure left below). Previous emissions (1-17 September 2014) travelled towards different directions (e.g. Atlantic Ocean, Arctic) and could have not been detected by the European Brewer and surface networks (see Figure to the right below). This has been clarified in the revised text.

The eruption was not a one day event. To clarify this we have added the following line in the captions of Figures 4, 5: "The arrow marks the starting date of the eruption (beginning on 31 August 2014 which continued to be active throughout the whole bimonthly period and beyond)."

[Figure]

Figure left: Cluster analysis of the HYSPLIT 120 hours forward trajectories from Bárðarbunga for the period 18-21 September 2014. Figure right: Cluster analysis of the HYSPLIT 120 hours forward trajectories from Bárðarbunga for the period 1-17 September 2014.

***Minor comment 6:*** *"Figure 9. The positive so2 signal at Taipei on 06/19 is compromised by the equivalent negative excursion on 07/17. Can the authors explain this?"*

**Answer to 6:** The positive SO2 signal at Taipei and the negative excursion are not connected to each other. The positive SO2 signal was due to the volcanic SO2 plume by Nabro as evidenced from the back trajectories analyses in figure 9.

*Minor comment 7: "12.38. "The increase in SO2 due to the passage of the Nabro volcano plume over the Canary Islands is significant using both methods, showing an offset between them (Figure 11)." What is meant by "showing an offset between them"? Do the different zeroing methods produce the offset, does the volcanic plume produce the offset, is the offset introduced to separate the lines, …?"*

**Answer to 7:** Both Brewer lines (data calibrated with two different methods at Izana) are free from offsets by subtracting from the data the non-perturbed pre-volcanic 10-day baseline. The text and Figure 11 have been revised.

*Minor comment 8: "Fig. 12 and 13.20. This figure has the same problem as Figs. 4 and 5. The mark indicating the eruption is misleading. First the text states that the eruption first occurred on 14 April. The figure legend claims the time frame is April-May so the reader assumes day 0 corresponds with April 1, but that would make the eruption occurring on 4 April. The text goes further to claim the eruption has three phases so it occurs on a number of days in April/May leaving the figure misleading again since it indicates an eruption on just one day. The authors could make this clear with appropriately labelled ordinates indicating what day 0 is, or better, use the actual dates of the month for simplicity, then add a mark on each day when there was significant effluent from Eyjafjallajökull."*

**Answer to 8:** The labels on the abscissa of Fig. 12 (and in all other figures) have been corrected to show the actual dates of the month as suggested. The arrow in Fig. 12 marks the beginning of the eruption of Eyjafjallajökull on 14 April.

*Minor comment 9: "13.36-14.5. It is surprising that back trajectories are not shown for Uccle and DeBilt, in fact no back trajectories are shown for Eyjafjallajökull, whereas a number of such trajectories were shown for Bárðarbunga. It appears that if there is no increase in so2 then the authors don't bother to calculate a back trajectory. That makes sense in most cases, but not when stations are as close as Uccle and DeBilt. The authors should consider adding back trajectories here to demonstrate that indeed one station was outside and one inside the plume on the same date. It would be a nice confirmation of the reliability of the back trajectories and the localized nature of the volcanic clouds."*

**Answer to comment 9:** This is a useful comment and in the revised text we replace lines 13.36-37 with the following section:

"We should note here that volcanic clouds can be rather narrow plumes with diameters on the order of a few tens kilometers (e.g. Stohl et al., 2011; Webley et al., 2012; Thorsteinsson et al., 2012; Kristiansen et al., 2012; Kokkalis et al., 2013) and thus it is possible that a detected volcanic layer at a specific station is not observed by neighbouring stations. The

measurements at Uccle and De Bilt that are located at a horizontal distance of 150 km are different during the Eyjafjallajökull episode and provide a formidable example. On 2 May 2010 the mean daily $SO_2$ is 5.8 DU at De Bilt and -0.2 DU at Uccle. As seen at the corresponding back trajectories for that day in Figure 13, air masses originating from Iceland arrive at De Bilt at heights 6-7 km and have been probably transporting the volcanic cloud over that station. In contrast similar back trajectories on the same day for the case of Uccle indicate transport of air masses from Iceland but at lower heights (3-4 km) that were probably not affected by the volcanic emissions. In another case of transport on 11 May 2010, Uccle was outside of the plume (see Figure 13) while the mean daily $SO_2$ for De Bilt was 0.9 DU. On 18 May 2010, both De Bilt and Uccle stations detected a volcanic cloud with $SO_2$ daily means of 1.7 DU and 1.2 DU respectively. To summarize, in spite the proximity of Uccle and De Bilt the transport heights and trajectories can have a different result in transporting volcanic gases."

The above response, new Fig. 13 (back trajectories for Eyjafjallajökull) and references have been added in the text as appropriate. The previous Fig. 13 (Kasatochi) now becomes Fig. 14.

[Figure]

Figure 13. HYSPLIT 120 hours back trajectories of air masses arriving at De Bilt (left column) and Uccle (right column) on 2 May 2010 (1st row), 11 May 2010 (2nd row), 18 May 2010 (3rd row).

*Minor comment 10: "14.10-19. The reader cannot compare any of these dates and details to Fig. 12 for the reasons stated concerning the timing of the eruption shown. Without that link then the value of these details are lost on the reader."*

**Answer to 10:** The dates on the X-axis of Fig. 12 have been corrected as before.

*Minor comment 11: "Sodankÿla should be Sodankylä everywhere it appears."*

**Answer to 11:** Corrected.

[revised manuscript text omitted]